# Adoptive macrophage directed photodynamic therapy of multidrug-resistant bacterial infection

Zehui Wang[1,3], Anhua Wu[2,3], Wen Cheng[2,3], Yuhe Li[2,3], Dingxuan Li[1], Lai Wang[1], Xinfu Zhang ◉[1] ✉ & Yi Xiao ◉[1]

Multidrug-resistant (MDR) bacteria cause severe clinical infections and a high mortality rate of over 40% in patients with immunodeficiencies. Therefore, more effective, broad-spectrum, and accurate treatment for severe cases of infection is urgently needed. Here, we present an adoptive transfer of macrophages loaded with a near-infrared photosensitizer (**Lyso700D**) in lysosomes to boost innate immunity and capture and eliminate bacteria through a photodynamic effect. In this design, the macrophages can track and capture bacteria into the lysosomes through innate immunity, thereby delivering the photosensitizer to the bacteria within a single lysosome, maximizing the photodynamic effect and minimizing the side effects. Our results demonstrate that this therapeutic strategy eliminated MDR *Staphylococcus aureus* (MRSA) and *Acinetobacter baumannii* (AB) efficiently and cured infected mice in both two models with 100% survival compared to 10% in the control groups. Promisingly, in a rat model of central nervous system bacterial infection, we performed the therapy using bone marrow-divided macrophages and implanted glass fiber to conduct light irradiation through the lumbar cistern. 100% of infected rats survived while none of the control group survived. Our work proposes an efaicient and safe strategy to cure MDR bacterial infections, which may benefit the future clinical treatment of infection.

Multidrug-resistant bacteria (MDR) are a leading cause of infection. They infect more than 25% of patients in Intensive Care Units (ICU) where they cause a high level of mortality[1–3]. Strong prophylactic procedures and a high level of clinical care are required in modern clinical medicine. Nevertheless, some infections are inevitable. The evolution of MDR bacteria combined with more invasive surgical procedures involving implanted medical devices has resulted in more severe invasive infections[4–7]. This is particularly so for immunodeficient patients such as those undergoing chemotherapy[8–10]. Immunodeficient patients have a high risk of developing severe bacterial infections leading to life-threatening sepsis. These patients are insensitive to recognize and respond to bacterial infections due to a lack of immune system activity, including decreased production of key immune cells, complement system function, and dysregulation or deficiency of inflammatory mediators. This increased risk is caused by several factors, including reduced production of antibodies, diminished phagocytic activities, and impaired T-cell functions. The ability to treat infections with antibiotics is sometimes ineffective due to antibiotic resistance and the inaccessibility of some sites of infection such as the meninges where the blood–brain barrier (BBB) hinders drug access[11–13]. This makes treatment of bacterial meningitis with antibiotics particularly difficult. In addition, administration of high

[1]State Key Laboratory of Fine Chemicals, Frontiers Science Center for Smart Materials Oriented Chemical Engineering, Dalian University of Technology, Dalian 116024, China. [2]Department of Neurosurgery, Shengjing Hospital of China Medical University, Shenyang 110055, China. [3]These authors contributed equally: Zehui Wang, Anhua Wu, Wen Cheng, Yuhe Li. ✉e-mail: zhangxinfu@dlut.edu.cn

doses of antibiotics over a prolonged period leads to serious side effects in immunodeficient patients[14,15]. So far, it remains an enormous challenge to cure multidrug-resistant bacterial infection in patients with immunodeficient status. Novel effective and broad-spectrum therapies as an alternative to antibiotics are sorely needed.

The ideal antibacterial strategy should be potent against a wide range of bacteria and specifically target them to maximize effectiveness while minimizing side effects. Photodynamic therapy displays high antibacterial efficiency and a wide spectrum of bacteria in vitro[16–20]. However, most photosensitizers of organic fluorophores lack targeting specificity to the live bacteria. So, in most cases, current trials have to in situ administrate a high dose of photosensitizers, such as spray on the wounds, to kill bacteria completely. And, inevitably, the over-dose of photosensitizer may damage the healthy tissue of the host if not properly dealt with. Some technologies under development can significantly improve the localization to infection sites, such as Qiao et al. reported the use of extracellular vesicles (EVs) to prolong blood circulation of nanomaterials. These strategies are currently more applicable to the epidermis or gastrointestinal tract bacterial infections, allowing for in situ drug delivery, and not to organs in vivo[21–25]. Limited targeting ability and therapy accuracy hinder its use for severe infections within host organs. The immune system, particularly macrophages, possesses remarkable targeting capabilities against pathogens and effectively eliminates bacteria through phagocytosis, making them efficient pathogen scavengers[21,26,27]. Unfortunately, patients with immunodeficiency exhibit impaired macrophages, leaving them vulnerable to bacterial infections due to insufficient and/or inactive immune cells. Furthermore, multidrug-resistant bacteria can evade immune cell recognition and clearance,

presenting an additional challenge for these patients. Despite attempts to preserve immune function through stimulation, infection-induced mortality rates remain high[21,26,28–30]. We previously suggested a novel method of enhancing innate immunity and eliminating MDR bacteria by employing adoptive transfer of macrophages that carry antimicrobial peptides[31]. The incorporation of antimicrobial peptides through mRNA technology further augmented the antibacterial effectiveness of macrophages. The innate targeting ability of macrophages towards bacteria accurately drew antimicrobial peptides (from mRNA expression) to bacteria in host mice and effectively eradicated bacteria in the circulatory system.

In this work, we propose an adoptive transfer of macrophages loaded with a photosensitizer in lysosome to boost innate immunity, and target and eliminate bacteria through photodynamic effect efficiently (Fig. 1a). To carry out this proposal, we first designed and synthesized a lysosome targeted near-infrared (NIR) photosensitizer (**Lyso700D**) that affords stably targeting ability to lysosome and superior photodynamic effect in the acidic lysosomal environment. This photosensitizer was then loaded into the lysosome of live macrophages through standard culturing. Then these "armed" macrophages were transferred into the host to capture live bacteria through innate immunity. When phagocytosed by these macrophages, bacteria were pulled to **Lyso700D** at a minimum distance within a single lysosome. Finally, **Lyso700D** killed MDR bacteria through the PDT effect and cured the infected host in two mice models with epidermal and central nervous system bacterial infection completely. Overall, as illustrated in Fig. 1a, this antibacterial strategy takes advantage of the accuracy of innate immunity and the efficiency of photodynamic therapy, kills MDR bacteria, and eradicates infection without injuring healthy tissue.

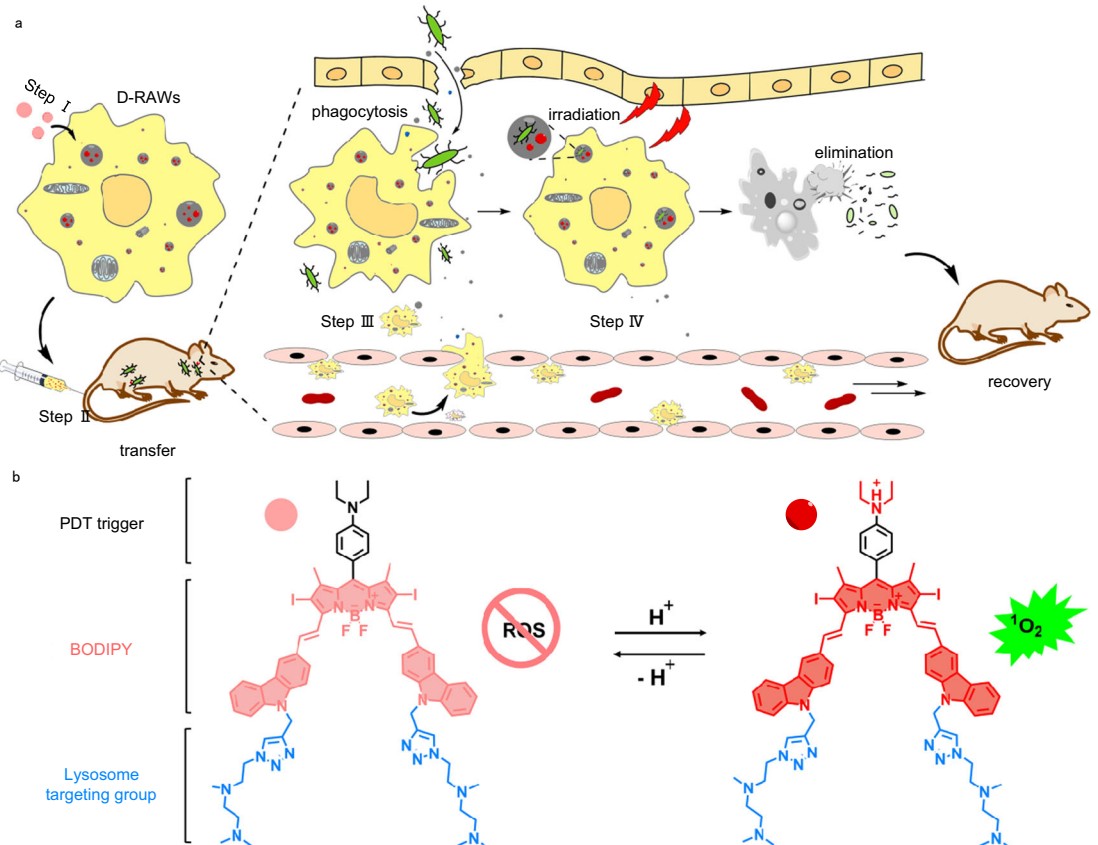

**Fig. 1 | Mechanisms of** Lyso700D **and D-RAWs in vivo. a** Schematic of adoptive macrophage-directed photodynamic therapy of bacterial infection in hosts. Step I, engineer D-RAWs by staining RAWs with **Lyso700D**; step II, transfer D-RAWs to infected mice through i.v. injection; step III, engineered D-RAWs track and capture bacteria; step IV, upon irradiation, D-RAWs self-destruct and kill the captured bacteria simultaneously. **b** Structure and mechanism of action of **Lyso700D**. Acidity turns up the PDT effect by restricting the photoinduced electron transfer process.

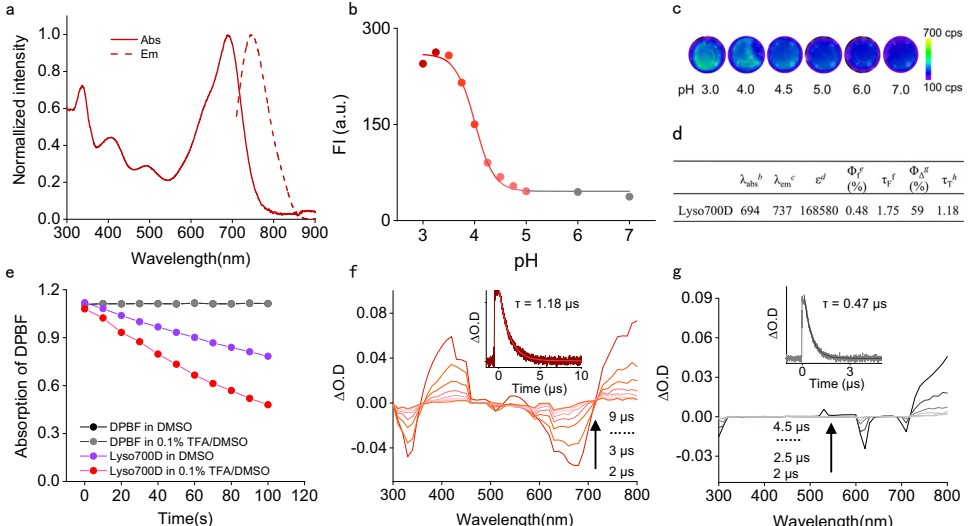

**Fig. 2 | Basic Photophysical Properties of** Lyso700D. **a** The normalized absorption and emission spectrums of **Lyso700D** in dimethyl sulfoxide (DMSO) with 0.1% trifluoroacetic acid (TFA). **b** Fluorescence intensity of **Lyso700D** at 700 nm against pH in Phosphate buffer saline (PBS). **c** Fluorescent images of **Lyso700D** aqueous solution versus different pH. **d** Basic photophysical properties of **Lyso700D**. [a] All the properties were tested in DCM; [b] maximum absorption wavelength, in nm; [c] maximum emission wavelength, in nm; [d] molar absorption coefficient, $M^{-1}\,cm^{-1}$; [e] absolute fluorescence quantum yield; [f] luminescence lifetimes, in ns; [g] singlet oxygen quantum yield with MB as standard ($\Phi_\Delta = 56\%$); [h] intrinsic triplet excited-state lifetime, in μs. **e** $^1O_2$ generation of **Lyso700D** trapped with DPBF in DMSO (with or without 0.1% TFA) irradiated with 740 nm LED light. **f** ns-TA spectra of **Lyso700D** in 0.1% TFA DCM after pulsed excitation, $\lambda_{ex} = 532$ nm. Inset: Decay trace of **Lyso700D** in 684 nm. **g** ns-TA spectra of **Lyso700D** in DCM after pulsed excitation, $\lambda_{ex} = 532$ nm. Inset: Decay trace of **Lyso700D** in 620 nm.

## Results and discussion

### Molecular design and synthesis

First, we design and synthesize this photosensitizer **Lyso700D** (Fig. 1b). Three features are considered to facilitate our proposal: stable lysosome targeting ability, high photodynamic efficiency, and strong NIR absorption. We employed 4, 4-difluoro-boradiazaindacene (BODIPY) as the core fluorophore, fabricated with two conjugated carbazole groups to extend its absorption wavelength to the NIR region, incorporated two iodine atoms to maximize the PDT effect, and grafted with two tertiary amine moieties to target lysosomes. Additionally, we introduced an extra diethylamine phenyl moiety to turn up the PDT effect in an acidic environment specifically through photoinduced electron transfer (PET). The synthetic route is illustrated in the supporting information (SI Fig. 1). The structure of **Lyso700D** has been characterized by $^1$H NMR and HRMS (SI Figs. 18–20).

### Basic photophysical properties of Lyso700D

Next, we measure the basic photophysical properties of **Lyso700D** to evaluate its compatibility for bio-application. As shown in Fig. 2a, **Lyso700D** exhibits intense absorption in the NIR region ranging from 500 to 800 nm with an extinction coefficient of 168580 $M^{-1}\,cm^{-1}$ in dichloromethane (DCM). This absorption feature indicates an efficient excitation at the NIR region for deep tissue. Meanwhile, **Lyso700D** also displays detectable emissions ranging from 710 to 900 nm that can be used for tracking the distribution of stained cells. Notably, this NIR emission turns down when pH is higher than 5.0 and turns up when pH is lower than 5.0 (Fig. 2b, c), due to the PET effect of the diethylamino phenyl moiety. Additionally, the fluorescence lifetime of **Lyso700D** in an acid environment is longer than in a non-acid environment (2.05 ns vs 1.58 ns) indicating the restriction of PET effect in acidic environments (SI Fig. 3c). This feature is in favor of maximizing the PDT therapy of **Lyso700D** within lysosomes while inhibiting the damage by PDT outside of lysosomes.

### Evaluation of $^1O_2$ generation in vitro under NIR light excitation

We also evaluate the singlet oxygen ($^1O_2$) generation efficiency by using 1, 3-diphenylisobenzofuran (DPBF) as a $^1O_2$ indicator. Absorption of DPBF at 418 nm decreased significantly within 100 s in the acidic pH solution under a low irradiance of 5 mW/cm$^2$ (0.05 J/cm$^2$, Fig. 2e). The singlet oxygen quantum yield was calculated to be 59%, even higher than that of methylene blue (MB, a popular photosensitizer used in antimicrobial and anti-tumor therapy, 56%, SI Fig. 3d, e). As a comparison, the singlet oxygen quantum yield dropped to 35% in the neutral solution, indicating that the PDT effect ($^1O_2$ generation efficiency) was pH-dependent and fully turned on only in the acidic pH solution. This pH response is in favor of maximizing its PDT effect in the lysosome (pH ~4.0) and restricting off-target injury outside of the lysosome. Further on, the $^1O_2$ generation efficiency under gradient irradiation fluence rate was investigated, which indicates an irradiance-dependent feature (SI Fig. 3f). Moreover, **Lyso700D** remained stable under irradiation (SI Fig. 3d), which is critical for biomedical applications.

To further investigate the activating mechanism of **Lyso700D** in low pH, the triplet excited state was measured by ns-TA spectroscopy (Fig. 2f, g). In particular, upon pulsed laser excitation at 532 nm, two negative bands centered at 330 nm and 684 nm were observed, which were the two ground-state bleaching bands (GSB). Meanwhile, by monitoring the decay of the GSB signal at 684 nm, the triplet excited state was found to be long-lived in a degassed solution (inset of Fig. 2f). Such a significant reduction in lifetime confirms the triplet state feature of the transient species. It was worth noting that there was a significant extension of the singlet state and triplet state under acidic conditions (inset of Fig. 2f, g). This is by reported results, that the PET process can cause fluorescence quenching and prohibit the occurrence of non-radiative transitions, thereby suppressing the generation of singlet oxygen[32,33].

### Targeting stability and photodynamic efficiency at the cellular level

Encouraged by the promising photophysical properties of **Lyso700D**, we then fully explore its photodynamic efficiency towards bacteria, using RAW264.7 cells (murine macrophages) to load this photosensitizer and capture bacteria. Commonly, bacteria will be phagocytosed by RAW264.7 cells into endosomes and then fuse with lysosomes through Toll-like receptors (TLRs), which makes lysosomes the perfect

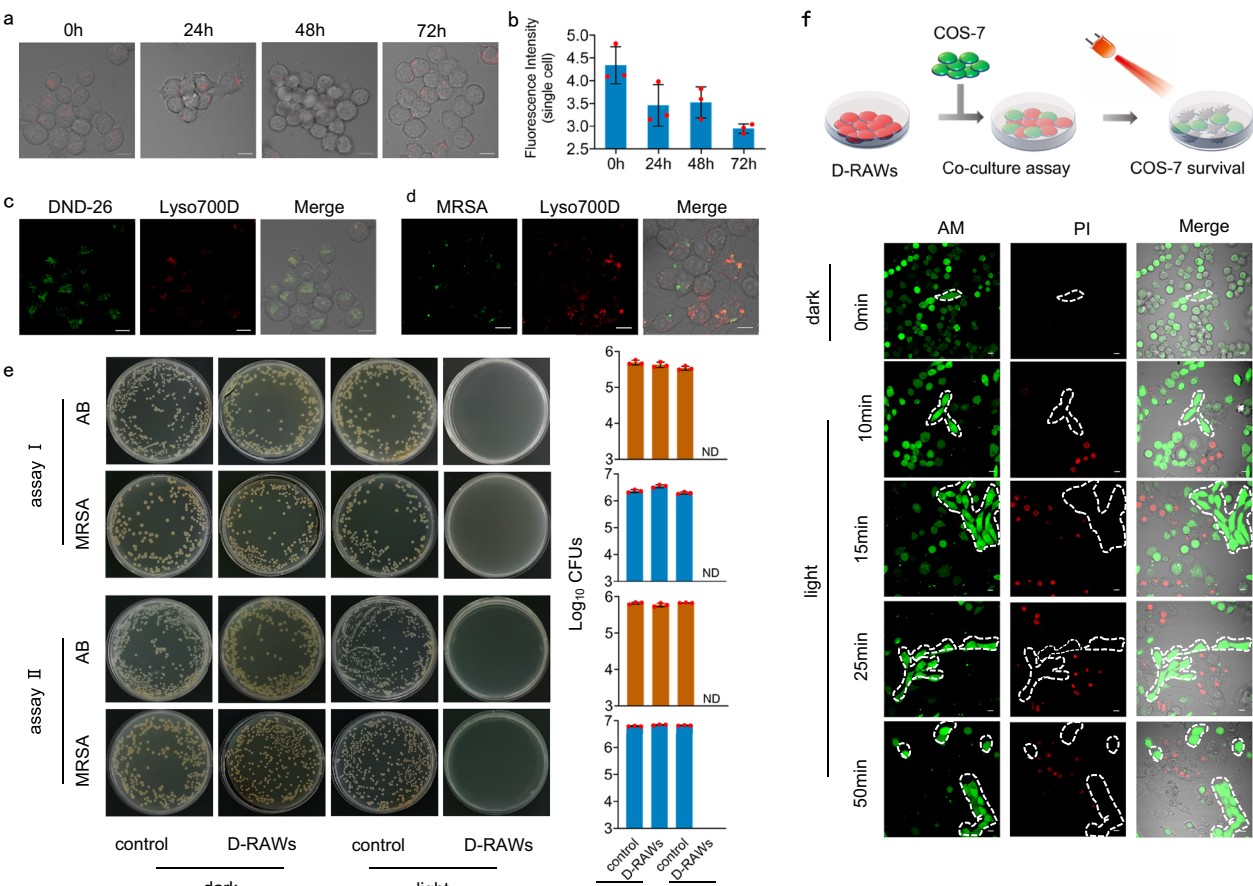

**Fig. 3 | Basal properties and antimicrobial properties of D-RAWs in vitro.**
**a** Confocal images of D-RAWs 0, 24, 48, and 72 h post staining with **Lyso700D**.
**b** Flow cytometry of D-RAWs 0, 24, 48, and 72 h post staining with **Lyso700D**.
**c** Confocal images of RAW264.7 cells co-stained with **Lyso700D** (red) and DND-26
(green). Each group of these experiments was performed on three independent
samples with similar results. **d** Confocal images of D-RAWs after taking MRSA
(stained with Mem-SQAC). Each group of these experiments was performed on
three independent samples with similar results. **e** Survival of AB and MRSA in
intracellular post phototherapy using freshly prepared D-RAWs (assay I) and
D-RAWs 72 h after preparation (assay II). **f** Safety assessment of phototherapy
through co-culture assay of D-RAWs and COS-7 cells. The viability of cells was
indicated through a Calcein-AM/PI co-staining assay. Each group of these experi-
ments was performed on three independent samples with similar results. Data in
**b** and **e** are from $n = 3$ biologically independent samples. All data in **b** and **e** are
presented as mean ± SD. ND: Not detected. Bar: 10 μm.

places to trap and kill bacteria. Therefore, we first confirmed the tar-
geting specificity and stability of **Lyso700D** towards lysosomes. We
performed colocalization imaging with LysoTracker® Green DND-26
(commercial lysosome tracker) through a confocal microscope. The
fluorescence distribution of **Lyso700D** in RAW264.7 cells showed high
a Pearson's correlation coefficient of 0.93 with DND-26, indicating
**Lyso700D** targets lysosomes specifically (Fig. 3c and SI Fig. 5). Further
on, we monitored the retention of **Lyso700D** in lysosomes over cell
proliferation through fluorescence imaging and flow cells and found a
stable lysosome targeting over 3 days (Fig. 3a, b). This lysosome tar-
geting stability is the foundation for accurate and efficient photo-
dynamic therapy.

We secondly confirmed the ability of RAW264.7 cells loaded with
**Lyso700D** (D-RAWs, prepared with standard method throughout this
study shown in supporting information) to capture and decompose
bacteria in vitro. We used gram-positive pathogenic bacteria and gram-
negative pathogenic bacteria to establish infection and demonstrate
the efficacy and versatility of D-RAWs. According to the advice from
doctors in the department of surgical operations and clinic, multidrug-
resistant *Staphylococcus aureus* (MRSA, ATCC 43300) is a typical gram-
positive pathogenic bacterium that leads to serious infection and is
difficult to eliminate due to multiple drug resistance. *Acinetobacter
baumannii* (AB, ATCC 19606) is one typical gram-negative and

opportunistic pathogenic bacterium in hospitals, especially in the ICU.
Both bacteria are highly related to serious infections such as menin-
gitis and sepsis. We incubated D-RAWs with the pathogenic bacteria
MRSA and AB respectively to conduct immune uptake. According to
fluorescence imaging, we observed an evident overlap of the fluores-
cence from MRSA (stained by membrane tracker Mem-SQAC) and
**Lyso700D**, which indicated that D-RAWs successfully take bacteria
into lysosomes in 0.5 h (Fig. 3d). We then treated these MRSA trapped
inside lysosomes with light irradiation, after removing the bacteria in a
culture medium using lysostaphin. Groups with light irradiation
showed complete elimination of two bacteria by counting bacterial
colonies (Fig. 3e, assay I). In comparison, control groups and groups
without light irradiation showed fast bacteria proliferation by counting
bacterial colonies. To confirm the durability of D-RAWs and **Lyso700D**
for practical use, we further incubated cultured D-RAWs (72 h posts the
initial stain with **Lyso700D**) with MRSA or AB and performed the same
light irradiation treatment. Both groups showed complete removal of
bacteria as well, indicating robust durability and broad-spectrum
bactericidal activity of these D-RAWs (Fig. 3e, assay II). To fully
demonstrate the advantages of our therapeutic strategy over the free
photosensitizer molecule, we also performed phototherapy on MRSA
using free **Lyso700D** alone. However, as shown in SI Fig. 7, there is no
difference in bacteria quantity between the control group and

**Lyso700D** groups (whenever with or without irradiation), which indicates the free **Lyso700D** is not able to kill bacteria effectively due to a lack of targeting specificity toward bacteria. It further demonstrates the significance of constructing a D-RAWS strategy that captures and delivers **Lyso700D** to bacteria through innate immunity. Specifically, D-RAWs were constructed by treating RAW264.7 cells with 250 nM of **Lyso700D**, yielding $2.56 \pm 0.03 \times 10^{-15}$ mole in each RAW264.7 cell average. Therefore, the equivalent concentration (molecules in each cell × cell numbers) of **Lyso700D** for therapeutic groups with D-RAWs in Fig. 3e is 130 nM, which reduces the dose of photosensitizer by over 94% compared to literature reports (ranging from 2.0 to 8.0 μM)[34–38]. The above comparison shows the therapeutic efficiency based on RAW264.7 cell is much higher than free photosensitizer molecules.

To verify the safety of this therapeutic strategy toward healthy cells, we performed a co-culture assay using D-RAWs and COS-7 cells (two cell lines can be distinguished through morphology features). Then we monitored the viability of D-RAWs and COS-7 cells post-light irradiation using live/dead cell co-staining assay. D-RAWs (rounded cells) showed gradual apoptotic as green fluorescence in the cytoplasm decreased and red fluorescence in the nucleus increased, by contrast, the neighboring COS-7 cells (fusiform cells marked broken line) remained alive as indicated by the intense green fluorescence in the cytoplasm (Fig. 3f). Even after light irradiation for 50 min (much higher than standard irradiation dose of 30 min), the COS-7 cells were still undamaged. Additionally, we confirmed the photodynamic effect of **Lyso700D** is directed by reactive oxygen species (ROS) through the standard ROS assay (SI Fig. 4c). We also confirmed that free **Lyso700D** affords excellent ROS generation ability and shows negligible dark toxicity toward RAWs through MTT assay and live/dead cell assay (SI Fig. 4a, b, d). As shown in SI Fig. 4a, at a staining concentration of 120 nM, only 30.38% Raw264.7 cells remain alive post 15 min under irradiation of 30 mW/cm$^2$ (27 J/cm$^2$), giving an IC$_{50}$ of 59.20 nM. Moreover, **Lyso700D** showed a significant photodynamic effect over tissue shelter of 10 mm, maintaining 60% efficiency to induce absorption decrease of DPBF and 30% efficiency to induce cell apoptosis, in a concentration- and irradiance-dependent manner through MTT assay (SI Fig. 6a–d). Generally, photodynamic therapy is efficient in destroying bacteria as well as healthy cells encountered. In our strategy, D-RAWs find and phagocytize bacteria into lysosomes, where Lyso-700D are loaded. The photodynamic effect can be not only applied to the bacteria but also confined within D-RAWs (as demonstrated in Fig. 3f). Therefore, our therapeutic strategy promotes the photodynamic effect on bacteria, while reducing the side effects to healthy tissue. The above results demonstrate that the photodynamic effect is highly efficient, yet the ROS generated by **Lyso700D** works within D-RAWs and thus is safe to host.

## Lyso700D for the treatment of epidermal bacterial infection in both immunodeficient and immunocompetent mice

Given the potent in vitro bactericidal activity of D-RAWs, we tested the ability of D-RAWs to track, capture, and eliminate bacteria in MRSA-induced epidermal infection in both immunodeficient and immunocompetent mice. The major goal of the epidermal model is to demonstrate the ability of D-RAWs to find the bacteria. In the epidermal model, bacteria accumulate in the superficial layer of the skin. If D-RAWs track and capture bacteria in the skin, it is feasible to demonstrate the "active tendency ability" of D-RAWs through in vivo fluorescence. Immunodeficiency refers to a state in which the immune system's functions are impaired, resulting in reduced immune cell activity and a weakened ability to fight infections. In an immunodeficient state, the number of immune cells, such as macrophages, T cells, and natural killer cells, may be reduced, compromising the body's defense mechanisms. Additionally, the chemotactic activity, which allows immune cells to migrate toward sites of infection, may be

diminished. Moreover, along with the reduced function of the complement system, the ability of immune cells to eliminate bacteria may also be compromised, leaving the body more susceptible to microbial invasions. Particularly, *S. aureus* can cause various skin and soft-tissue infections that may develop into sepsis and lead to death in the host with immunosuppression. And the prevalence of drug resistance makes such infection highly lethality. We first treated healthy mice with cyclophosphamide (CY) for three consecutive days to induce an immunocompromised state with low white blood cells (WBCs), lymphocytes (LYMs), mononuclear cells (MONs) and neutrophile granulocytes (GRANs) (Fig. 4f, g, SI Fig. 8). See "Construction of immunodeficiency model in mice and rats" in the supporting information for detailed characterization. We then built epidermal infection by subcutaneous injection of MRSA into the wound on their right legs. We also built a common wound without injection of bacteria on the left legs as a comparison (Fig. 4a). After epidermal infection formed, mice were injected with PBS, RAWs (PBS-RAWs), **Lyso700D** (D), and D-RAWs intravenously (i.v./i.v + i.p.). Following the principle of intravenous administration, there is a limitation of dose for each administration on the mice model (2.5 mL/kg mice). Generally, after i.p. inoculation, D-RAWs are transported by the lymphatic system to blood within a short period and then distributed to other organs via blood circulation[39,40]. Administration of i.p. + i.v. may contribute to eliminating the bacteria in both the peritoneal cavity and blood, thus maximizing efficacy. Therefore, we performed an administration of D-RAWs with combined intraperitoneal and intravenous injections. The total injection dose of D-RAWs in the i.v. + i.p. group (i.p., $1 \times 10^6$ cells + i.v., $1 \times 10^6$ cells) is equal to that of the i.v. group ($2 \times 10^6$ cells). Twelve hours after the injection, we found a significant fluorescence signal from **Lyso700D** at the infection sites for two D-RAWs groups, which lasted for at least 96 h with peak intensity between 24–72 h (Fig. 4b and SI Fig. 11a, c). No significant fluorescence was observed from the common wound on the left leg (Fig. 4b). By contrast, mice in D groups showed no fluorescence signal either at the inflammatory site on the right leg or the common wound on the left leg (Fig. 4b and SI Fig. 11b, d). The fluorescence signal in D groups was mainly enriched in the spleen and liver and diminished in three days (Fig. 4b and SI Fig. 11). We also evaluated the distribution feature of D-RAWs and **Lyso700D** in a shorter time. As shown in SI Fig. 9, there is almost no fluorescence on the inflammation site of the right leg or the common wound of the left leg at 1, 3, and 6 h in both groups. These results fully confirm the ability of D-RAWs to track bacteria/ infection in vivo. Whereas, **Lyso700D** showed no specific tendency to the infection site in the time range of 1–72 h after the administration. It also indicates a broad photodynamic treatment window between 12–96 h post injections of D-RAWs. Additionally, we observed significant fluorescence in the main organs through ex vivo imaging (SI Fig. 10). Considering the longer retention characteristics of D-RAWs than Lyso700D in the organs (SI Fig 11), we can boldly rule out the cause of the leakage of free dye Lyso700D. We infer that fluorescence in the main organs has resulted from two potential reasons. The first one is, according to the distribution feature of macrophages in the literature, they widely distribute throughout an organism's organs and tissues, ensuring immune surveillance and tissue homeostasis. The second one is the chemotactic gradient drives macrophages specifically to the site of bacterial infection. Here, only a number of macrophages, other than all macrophages, are deployed to fight against bacteria, since our immune system is conservative[41,42], which means there are still plenty of macrophages sticking to their original location. According to the CFU of organ samples (Fig. 4c and SI Fig. 12), the epidermal bacterial infection in the PBS-treated immunosuppressed mice developed into bacteremia 36 h after the bacteria injection. D-RAWs, therefore, tracked, captured, and eliminated these bacteria in organs in the treatment group. According to the CFU of organ samples (Fig. 4c and SI Fig. 12), the epidermal bacterial infection in the PBS-treated immunosuppressed mice developed into

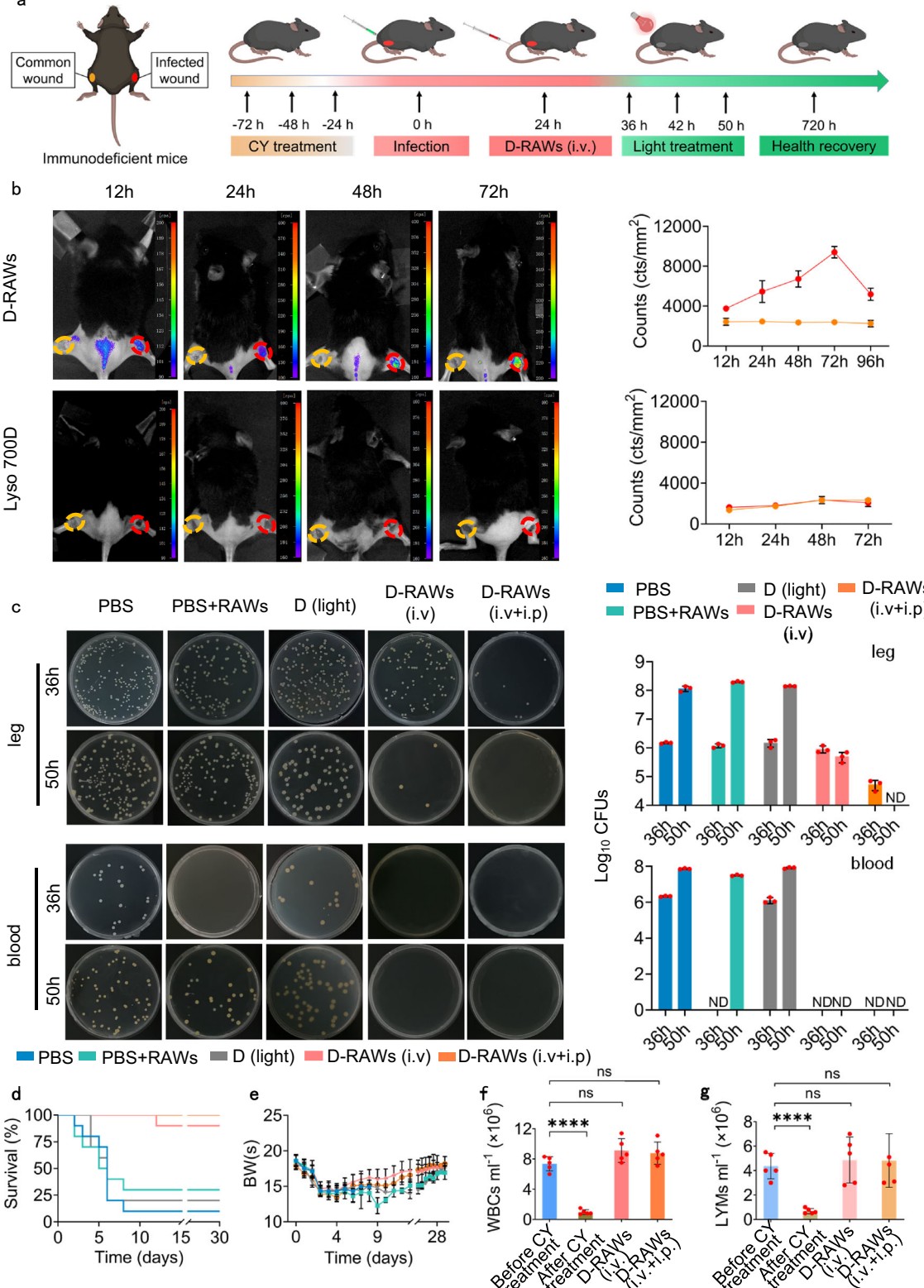

**Fig. 4 | D-RAWs for the treatment of epidermal bacterial infection in immunodeficient mice. a** Schematic illustration of the construction of epidermal bacterial infection in immunodeficient mice and the therapy routine based on D-RAWs (C57BL/6 mice were i.p. injected with cyclophosphamide at a dosage of 100 mg kg$^{-1}$ for three consecutive days before bacterial infection.) (Created with BioRender.com). **b** Fluorescence images of infected mice post injecting D-RAWs or **Lyso700D**. **c** Bacterial burden in the infection sites and blood at 36 h and 50 h post phototherapy. **d** The survival rate of mice with epidermal bacterial infection post phototherapy. **e–g** The BWs (**e**), WBCs (**f**), and LYMs (**g**) of mice. Data in **b** and **c** are from $n = 3$ biologically independent samples. Data in **d** and **e** are from $n = 10$ biologically independent samples. Data in **f** and **g** are from $n = 5$ biologically independent samples. All data in **b–g** are presented as mean ± SD. Statistical analysis was performed by a two-tailed unpaired $t$-test (**f, g**). ns no significant difference; *$P < 0.05$; **$P < 0.01$; ***$P < 0.001$. **d, e** Share a set of color codes. PBS (blue); PBS + RAWs (green); D (light) (gray); D + RAWs (i.v.) (pink); D + RAWs (i.v. + i.p.) (orange). ND not detected, ns non-significant.

bacteremia at 36 h after the bacteria injection. D-RAWs, therefore, tracked, captured, and eliminated these bacteria in organs in the treatment group.

We, therefore, treated the infection site with 50 mW/cm$^2$ (90 J/cm$^2$) light irradiation (the irradiance is much lower than the safety threshold of 300 mW/cm$^2$) for 30 min/time/day for two successive days (a standard method of irradiance in this study, detailed information is shown in supporting information). We found significant elimination of bacterial CFU in infected legs in both D-RAWs (i.v.) and D-RAWs (i.p. + i.v.) groups, which avoided the transfer of MRSA to blood and finally avoided the death of mice. In particular, D-RAWs (i.p. + i.v.) completely removed bacteria from the infection site and blood system after treatments. D-RAWs (i.v.) removed more than 99% of bacteria at the injection site after treatment and also avoided bacteria transfer to the blood system (Fig. 4c, SI Fig. 12). By contrast, for the control group (PBS), MRSA proliferated rapidly and transferred to the blood system, and led to a rapid decline in body weight of all mice and 90% of death in 8 days (Fig. 4d, e) due to the development of sepsis (SI Fig. 12). For the PBS-RAWs group, the treatment delayed the development of sepsis initially as indicated by lower CFU in the blood system (0 CFU at 36 h, $3.2 \pm 0.26 \times 10^7$ CFU at 50 h) compared with PBS group ($2.17 \pm 0.15 \times 10^6$ CFU at 36 h, $7.4 \pm 0.53 \times 10^7$ CFU at 50 h) (Fig. 4c). This result indicates that the active immune cells in the immunocompetent mice or the supplemented immune cells in the immunodeficiency mice (such as the PBS-RAWs group in our assay) can fight bacterial infections initially to some extent; it also reflects the vulnerability of immunodeficient mice against bacterial infections. However, PBS-RAWs failed to eliminate MRSA and left a death up to 60% (Fig. 4c–e, SI Fig. 12), which indicated native RAWs provide limited efficiency in treating MRSA invasion, which still needs to be accompanied by more effective treatments. For the **Lyso700D** group, the treatment failed to inhibit MRSA proliferation either in the blood or at the infection site and left the death of 95% of mice (Fig. 4c–e, SI Fig. 12), which by contrast highlighted the necessity of targeting bacteria using macrophages. These results demonstrate the in vivo bactericidal activity of photodynamic therapy based on D-RAWs. Additionally, both D-RAWs (i.v.) and D-RAWs (i.p. + i.v.) groups show significant potency demonstrating these two regular administration routes are compatible with our therapy strategy. 30 days after the infection, there was one surviving mouse in the control. However, bacterial spores were detected in the scar of the surviving mouse (SI Fig. 13), indicating a probability of infection recurrence. As a comparison, no bacteria were detected from the healed mice in the D-RAWs group (SI Fig. 13). And these mice showed a complete return to a healthy weight (BWs), white blood cells (WBCs), and LYMs levels after 30 days (Fig. 4e–g). Importantly, according to our calculation, the equivalent dose (weight of **Lyso700D** in each cell × cell numbers/weight of mice) is 0.32 mg/kg mice, which is relatively low doses of photosensitizer in recently reported treatment of epidermal bacterial infection (ranging from 8.53 μg/kg to 24 mg/kg mice, SI Table 2)[23,37,38,43–46]. These results support our proposal: accurate delivery of a limited amount of photosensitizer to approach bacteria at the single cell level and efficient elimination of bacteria through phototherapy.

To fully demonstrate the efficacy of the treatment in immunocompetent hosts, specifically the competition with natural macrophages, we performed an epidermal bacterial infection in immunocompetent mice. We built epidermal infection by subcutaneous injection of MRSA into the wound on their right legs. We also built a common wound (without injection of bacteria) on the left leg as a comparison. After epidermal infection formed, mice were injected with PBS, and D-RAWs intravenously (i.v.) (Fig. 5a). For D-RAWs groups, we found a significant fluorescence signal from **Lyso700D** at the infection sites on the right legs 12 h after the injection, but no fluorescence from the common wound on the left legs (Fig. 5b, SI Fig. 15b). This result confirms the ability of D-RAWs to track

and capture bacteria in immunocompetent mice. We then treated the infection site with light irradiation at an irradiance of 50 mW/cm$^2$ (90 J/cm$^2$) light irradiation for 30 min/time/day for two successive days (a standard method of irradiance in this study, detailed information is shown in supporting information, Fig. 5a). We found complete elimination of bacterial CFU in infected legs, blood systems, and organs in D-RAWs groups (Fig. 5c, SI Fig. 14), and rapid healing of the infected wounds in only 7 days (SI Fig. 15a). The survival rate was 100% in 30 days, and all mice recovered to normal levels of BWs, WBCs and LYMs (Fig. 5d–f, SI Fig. 15c). By contrast, for the control group, MRSA proliferated rapidly on the infection sites and transferred to the blood system (Fig. 5c). However, due to the defense mechanism of the competent immune system, the proliferation of MRSA was inhibited to a low, but detectable, CFU in the main organs of the immunocompetent mice (SI Fig. 14). All these mice survived without any treatment but formed serious scars and maintained bacterial endospore on the wounds 30 days after the infection (Fig. 5g, SI Fig. 15a, c). They all recovered to normal weight and levels of BWs, WBCs, and LYMs (Fig. 5d–f) in 30 days. As a comparison, 90% of infected immunodeficient mice succumbed when no treatments were provided (color code: Blue; Fig. 4d). Furthermore, hematoxylin–eosin (H&E) staining (SI Fig. 15d) was performed to evaluate pathological organ changes in these "recovered" immunocompetent mice. For mice in PBS groups, severe vacuolation in the liver, heterocyclic nuclei (yellow arrow), and severe extramedullary hematopoiesis (red arrow) in the splenic parenchyma were observed. The structure of lung tissue was disordered with diffuse red blood cell infiltration (blue arrow), and the ischemia of renal tissue led to the fragmentation of acute tubular necrosis (green arrow). For mice in the D-RAWs group, no pathological damage was observed in all organs. These results prove that the D-RAWs provide important supplementary to immune defense and accelerate the recovery of the infected host when the natural macrophages are still robust.

## Treatment of meningitis in immunodeficient mice with Lyso700D

The promising in vivo therapeutic results in the epidermal bacterial infection model encourage us to evaluate antimicrobial activity in the meningitis model. This is a tangled clinical situation as the bacteria is within the cerebrospinal fluid (CSF). This infection can be severe and fatal, leading to cerebral edema, focal neurologic deficits and even sepsis, multi-organ failure, and death of patients in immunodeficient states[47–49]. The administration of antibiotics is the standard clinical therapy, yet they are not efficient against meningitis due to the limitation of the BBB and poor pharmacokinetics in critical patients[50,51]. Therefore, in addition to the acute results, cured patients will experience long-term sequelae, such as cognitive impairment, hearing loss, and motor deficits, due to brain nerve damage. To eliminate these bacteria, drugs or other therapies should pass the BBB. We built the meningitis models using two pathogenic bacteria, MRSA and AB, respectively (Fig. 6a) to simulate two typical and fatal infections. We first treated healthy mice with cyclophosphamide (CY) to induce an immunocompromised state and then injected AB or MRSA into CSF in mice through standard brain surgery. The development of meningitis was quite fast and led to the death of mice within 2 days and 100% of deaths within only 7 days (Fig. 6e, f). To evaluate the efficiency of our therapeutic strategy for meningitis, we administered D-RAWs to infected mice via intravenous injection (i.v.) and then conducted phototherapy. Firstly, to confirm the biodistribution of D-RAWs through intravenous injection (i.v.), we injected D-RAWs into immunodeficient mice with and without meningitis, harvested CSF for flow-fluorescence cell sorting and fluorescence imaging (prepared with standard method throughout this study shown in supporting information). We identified the presence of D-RAWs (4350 cells/ml) in the CSF of meningitis mice by detecting the fluorescence from **Lyso700D**

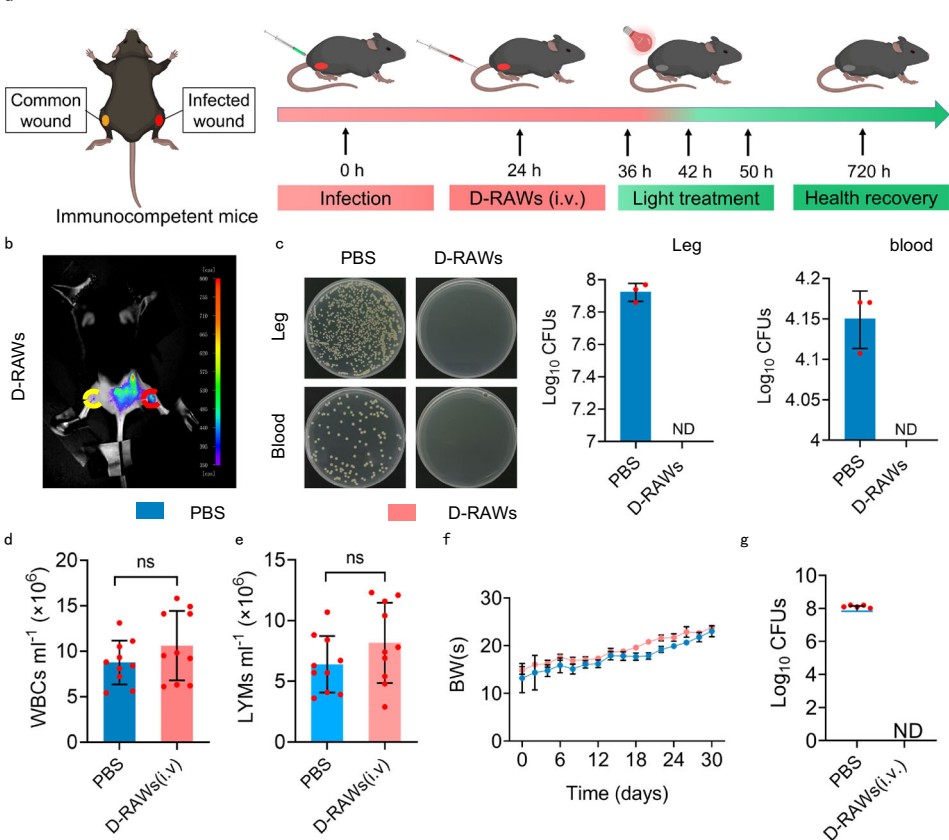

**Fig. 5 | D-RAWs for the treatment of epidermal bacterial infection in immunocompetent mice. a** Schematic illustration of the construction of epidermal bacterial infection in immunocompetent mice and the therapy routine based on D-RAWs (Created with BioRender.com). **b** Fluorescence images of infected mice post 12 h injecting D-RAWs. **c** Bacterial burden in the infection sites and blood at 50 h post phototherapy. **d**–**f** The WBCs (**d**), LYMs (**e**), and BWs (**f**) of mice after 30 days. **g** Bacterial burden in the leg of each surviving mouse by PBS and D-RAWs after 30 days. Data in **b** and **c** are from n = 3 biologically independent samples. Data in **d**–**g** are from *n* = 10 biologically independent samples. All data in **c**–**g** are presented as mean ± SD. Statistical analysis was performed by a two-tailed unpaired *t*-test (**d**, **e**). ns no significant difference; *\*P* < 0.05; *\*\*P* < 0.01; *\*\*\*P* < 0.001. **c**–**g** Share a set of color codes. PBS (blue); D-RAWs (pink). ND not detected, ns non-significant.

in RAWs (Fig. 6b). As a comparison, no cells were observed in none meningitis mice with or without D-RAWs injection (Fig. 6b). This is consistent with the immune privilege state of brain and provides strong support to demonstrate the intelligent targeting ability of D-RAWs towards bacteria/meningitis.

Secondly, we treated the infected mice in two D-RAWs groups over intact brain bone with light irradiation at an irradiance of 50 Mw/cm² (90 J/cm²) light irradiation for 30 min/time/day for two successive days (a standard method of irradiance in this study, detailed information is shown in supporting information). We found a complete elimination of AB and MRSA in the brain right after phototherapy by harvesting the brains of both PBS and D-RAWs groups (Fig. 6c, d). Moreover, 100% of mice in the therapy groups survived and recovered from the infection (Fig. 6e, f). Further on, all cured mice recovered from the immunosuppressed state in 30 days as indicated by normal levels of WBCs and LYMs (Fig. 6g, h). In another 20 days, we harvested the brains and main organs of these cured mice for detailed analysis. No bacteria were found in the brains or main organs (heart, liver, spleen, lungs, and kidneys) and the levels of BWs, WBCs, and LYMs recovered to the normal state (Fig. 6f–h). In this therapy, according to our calculation, the equivalent dose (weight of **Lyso700D** in each cell × cell numbers/weight of mice) is 0.016 mg/kg mice, which is only 5% of the dose used in epidermal infection in our study. H&E staining of the brain demonstrated no significant injury due to the therapy, suggesting the safety of our treatment (Fig. 6i). These results fully demonstrate the advantages of using D-RAWs to treat bacterial infections: high efficiency due to targeting specificity and minimum side effect due to low dose of the photosensitizer.

Lastly, we performed a rat model to demonstrate the efficacy of our therapeutic strategy and its potential to transfer to humans. Cell resource and irradiation depth are two major hindrances to transferring the therapy to humans. In this experiment, we used primary bone marrow-divided macrophages (BMDMs) as the cell resource, which can be translatable for future clinical applications. We used implanted glass fiber to conduct light irradiation through the lumbar cistern, which is a regular clinical operation, to overcome the limitation of irradiation depth. The rationale of this irradiation is that the rats generate 120–300 μL of CSF per day, which circulates in the spine at the speed of 10–40 μl/min. This circulation of the CSF promotes the movement of macrophages within the CSF, which moves the macrophages passing through the irradiation area multiple times during the phototherapy and leads to the elimination of bacteria. Specifically, we first treated healthy rats with cyclophosphamide (CY) to induce an immunocompromised state and then injected MRSA into their CSF through brain surgery to build meningitis (Fig. 7a). At the same time, we generated BMDMs from rat bone marrow and loaded these BMDMs with **Lyso700D** (D-BMDMs). The development of meningitis was fast and led to the death of rats within 36 h and 100% fatality within 6 days in two groups (rats in both groups were treated with the same phototherapy through implanted optical fibers) injected with either PBS or unstained BMDM (Fig. 7g). In the therapeutic group, we first administrated D-BMDMs to infected rats

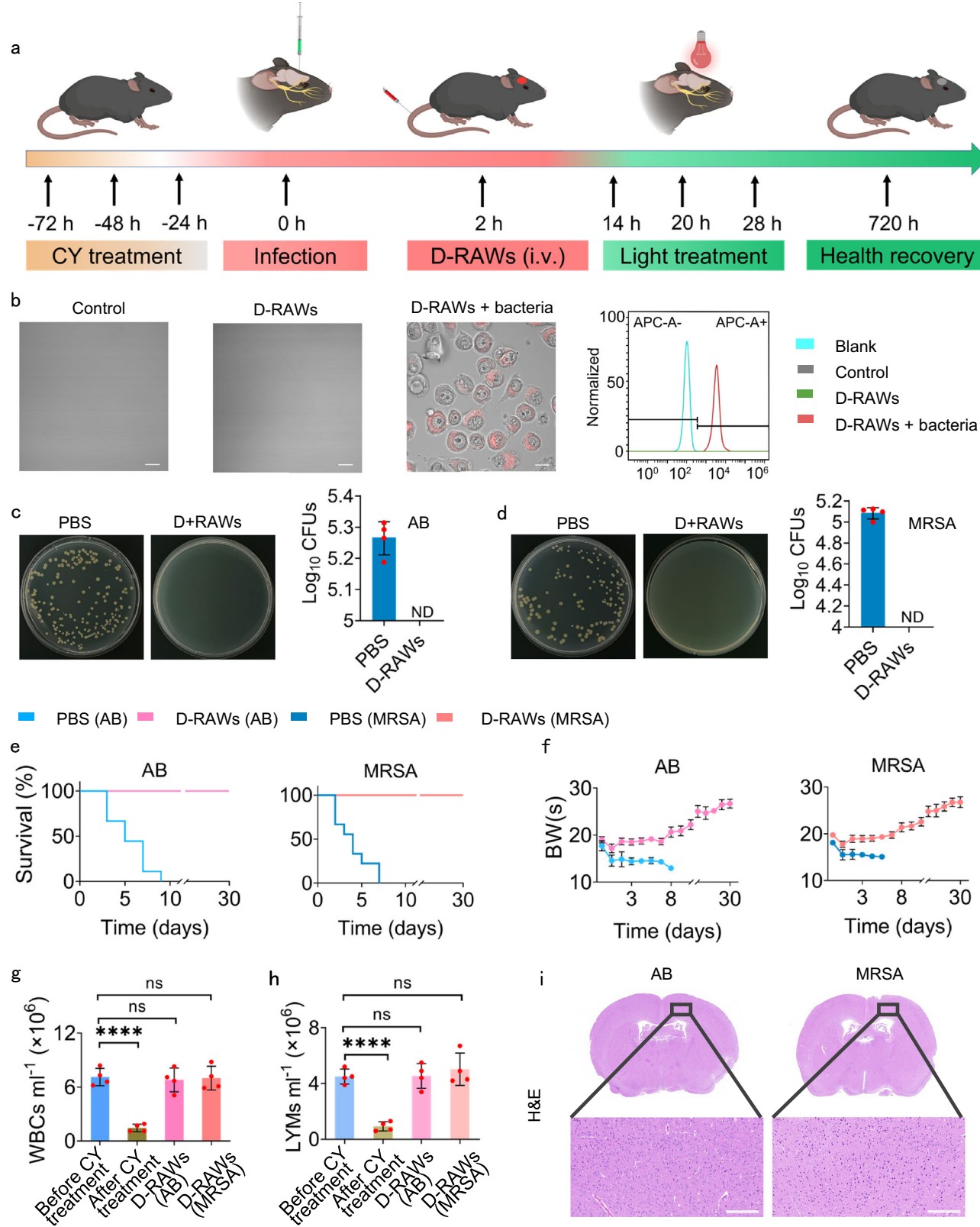

via intravenous injection (i.v.) and then conducted phototherapy through implanted glass fiber at the lumbar cistern (Fig. 7a, b). After one phototherapy (30 min, 80 mW/cm², 144 J/cm², LED light with a wavelength of 671 nm), all rats survived and lived beyond the experiment time of 30 days. Most importantly, there were no detectable bacteria in the brain and blood soon after the phototherapy (Fig. 7c, g, SI Figs. 16 and 17). These treated rats recovered

from the immunosuppressed state in 30 days as indicated by normal levels of BWs, WBCs, and LYMs (Fig. 7d, e, g). After another 20 days, we harvested the brains and main organs of these cured rats and found no bacteria in the brains or main organs (SI Fig. 17b). Our strategy shows a high anti-bacterial efficiency using BMDMs and clinical procedure, demonstrating its potential to transfer to humans. For future transfer, induced pluripotent stem cell technologies may

**Fig. 6 | D-RAWs for the treatment of meningitis bacterial infection in immunodeficient mice. a** Schematic illustration of the construction of meningitis in immunodeficient mice model and the therapy routine based on D-RAWs (Created with BioRender.com). **b** Confocal images and flow-fluorescence cell sorting of CSF from healthy and infected mice with or without injecting D-RAWs. Bar: 10 μm. **c**, **d** AB (**c**) and MRSA (**d**) burden in the brain right after the third phototherapy. **e** The survival rate of mice with CSF bacterial infection post phototherapy. **f–h** The BWs (**f**), WBCs (**g**), and LYMs (**h**) of mice. **i** H&E staining of brain slices of cured mice 30 days after the phototherapy. Bar: 200 μm. Each group of these experiments was

performed on 3 independent samples with similar results. Data in **b** is from $n = 5$ biologically independent samples. Data in **c** and **d** are from $n = 4$ biologically independent samples. Data in **e** and **f** are from $n = 9$ biologically independent samples. Data in **g** and **h** were from $n = 4$ biologically independent samples. All data in **c–h** are presented as mean ± SD. Statistical analysis was performed by a two-tailed unpaired $t$-test (**g**, **h**). ns no significant difference; $^*P < 0.05$; $^{**}P < 0.01$; $^{***}P < 0.001$. **c–f** share a set of color codes. PBS (AB) (blue); D-RAWs (AB) (rose red); PBS (MRSA) (dark blue); D-RAWs (MRSA) (pink). ND not detected, ns non-significant.

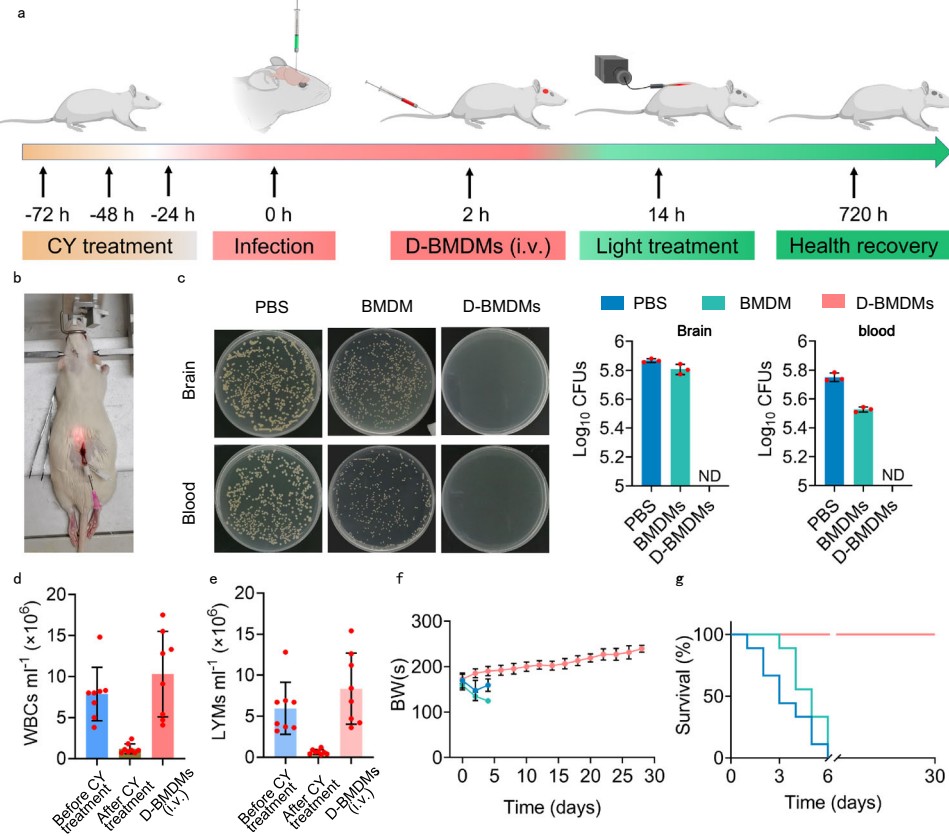

**Fig. 7 | D-RAWs for the treatment of meningitis bacterial infection in immunodeficient rats. a** Schematic illustration of the construction of meningitis in immunodeficient rats' model and the therapy routine based on D-BMDMs (Created with BioRender.com). **b** The real picture of light irradiation through the lumbar cistern by implanting glass fiber. **c** MRSA load in the brain and blood in the immediate post-phototherapy period. **d–f** The WBCs (**d**), LYMs (**e**), and BWs (**f**) of

rats. **g** The survival rate of mice with CSF bacterial infection post phototherapy. Data in **c** was from $n = 3$ biologically independent samples. Data in **d–g** were from $n = 9$ biologically independent samples. All data in **c–f** are presented as mean ± SD. (**c**, **f**, and **g**). Share a set of color codes. PBS (blue); BMDM (green); D-BMDMs (pink). ND not detected, ns non-significant.

be a solution to cell resource[52], and sound-sensitive dyes may be a solution to conduct deep-tissue dynamic therapy[53–56].

In conclusion, we propose an alternative antibacterial concept by taking advantage of the bacteria-chemotaxis and phagocytosis of macrophages (innate immunity) and the broad-spectrum antibacterial efficiency of PDT. Our results prove that these adoptive macrophages can supplement innate immunity, capture bacteria into lysosomes, eliminate bacteria through PDT, and cure the infected mice at 100%. Our therapy displays advantages over current PDT strategies in efficiency, bio-safety, and clinical operability. The dose of photosensitizer is much lower than reported data and the potent PDT on bacteria is restricted within single adoptive macrophages, which promotes the safety of our therapeutic strategy. Our strategy aims to treat severe conditional infection in immunodeficient hosts, such as patients in the ICU. We demonstrated its efficacy and potential to transfer to humans by a rat model, using bone marrow-divided macrophages as cell resources and a lumbar cistern to conduct light irradiation to the deep

site. Future work will focus on optimizing the dose of photosensitizer and macrophages. Generally, this strategy may provide an essential chance for bacterial infection in immunodeficient patients.

## Methods

### Ethics statement

Our research complies with all relevant ethical regulations. All animal studies were performed following ARRIVE guidelines. All animal experiments, including mice and rats were approved by the animal research ethics committee of Dalian University of Technology (DUT20210902). The animal study complied with relevant ethical regulations for animal testing and research.

### Materials and instruments

All chemicals were purchased from commercial suppliers (Bidepharm or Energy Chemical) and used without further purification. Hoechst 33342, LysoTracker® Green DND-26, 2′, 7′-dichlorodihydrofluorescein

diacetate (DCFH-DA), Calcein-AM/PI Double Stain Kit, and DND-26 were purchased from Yeasen Biotech Co., Ltd. 3-(4,5-dimethylthiazol-2-yl)-2,5-diphenyl-2h-tetrazoliubromide (MTT) was purchased from Bidepharm. The chicken breast was purchased from Walmart supermarket.

$^1$H-NMR was measured in d6-DMSO, CDCl$_3$ with TMS as an internal reference. The 400, 500 MHz $^1$H-NMR. Mass spectra were measured with an Agilent LC/Q-TOF MS.

UV-vis-NIR spectra were performed on Agilent 8453 UV-visible or Lambda 750 S spectroscopy system with a path length of 1 cm. Fluorescence images were obtained with a confocal laser scanning microscope (Olympus Fluoview FV1000). Small animals' fluorescence imaging was carried out by the NightOWL II LB983 living imaging system.

### Synthesis of compound 5
**Compound 3** (500 mg, 772.68 μmol) and **compound 4** (369 mg, 1.7 mmol) were dissolved in 10 mL of anhydrous toluene, and 0.5 mL of piperidine and 0.5 mL of glacial acetic acid were added successively. The mixture was heated to Reflux at boiling point. The progress of the reaction was monitored by TLC. The solution was diluted with deionized water and extracted with CH$_2$Cl$_2$ (3×). The combined organic solvent was dried with MgSO$_4$ and concentrated. The crude product was purified by flash chromatography (silica gel) to afford a green solid **compound 5** (0.587 g, 554.73 μmol) in a 70.50% yield. $^1$H NMR (400 MHz, DMSO) δ 8.47 (s, 2H), 8.39–8.28 (m, 4H), 7.89–7.78 (m, 4H), 7.70 (dd, J = 24.9, 12.4 Hz, 4H), 7.54 (t, J = 7.7 Hz, 2H), 7.28 (d, J = 7.5 Hz, 2H), 7.15 (s, 2H), 6.86 (d, J = 8.4 Hz, 2H), 5.36 (d, J = 1.7 Hz, 4H), 3.43 (d, J = 6.8 Hz, 4H), 3.29 (s, 2H), 1.61 (s, 6H), 1.16 (s, 6H). HRMS (ESI, m/z): [M + H]$^+$ calcd for C$_{55}$H$_{45}$BF$_2$I$_2$N$_5$, 1078.1820; found, 1078.1810. [M+Na]$^+$ calcd for C$_{55}$H$_{44}$BF$_2$I$_2$N$_5$Na, 1100.1639; found, 1100.1624. [M + K]$^+$ calcd for C$_{55}$H$_{44}$BF$_2$I$_2$N$_5$K, 1116.1379; found, 1116.1367.

### Synthesis of Lyso700D
**Compound 5** (25 mg, 23.20 μmol), **compound 6** (4.77 mg, 2.84 μmol), sodium ascorbate (0.41 mg, 2.32 μmol), CuSO$_4$·5H$_2$O (0.053 mg, 2.32 μmol) were added to a mixture of THF and H$_2$O (v/v 3: 1) under argon protection, stirred at room temperature for 24 h. The solvent was distilled off under reduced pressure and the residues were purified by silica gel column chromatography by solvent CH$_2$Cl$_2$: EtOH (35: 1, v/v) to obtain the dark green solid **Lyso700D** (22 mg, 16.20 μmol) in a 69.81% yield. $^1$H NMR (400 MHz, CDCl$_3$) δ 8.25 (d, J = 14.9 Hz, 4H), 8.16–8.05 (m, 4H), 7.91–7.74 (m, 6H), 7.65 (d, J = 16.6 Hz, 2H), 7.53 (t, J = 7.5 Hz, 2H), 7.23 (d, J = 7.4 Hz, 2H), 7.05 (d, J = 8.4 Hz, 2H), 6.79 (d, J = 7.9 Hz, 2H), 5.62 (s, 4H), 4.38 (s, 4H), 3.45 (d, J = 7.1 Hz, 4H), 2.71 (s, 4H), 2.54 (s, 8H), 2.11 (s, 12H), 2.01 (s, 6H), 1.31 (s, 6H), 1.28–1.19 (m, 6H). HRMS (ESI, m/z): [M + H]$^+$ calcd for C$_{69}$H$_{79}$BF$_2$I$_2$N$_{15}$, 1420.4788; found, 1420.4747. [M + 2H]$^{2+}$ calcd for C$_{69}$H$_{80}$BF$_2$I$_2$N$_{15}$, 710.7430; found, 710.7410.

### In vitro singlet oxygen ($^1$O$_2$) detection of Lyso700D
The $^1$O$_2$ of **Lyso700D** was assessed by using 1, 3-diphenylisobenzofuran (DPBF). The absorbance of DPBF at 415 nm was adjusted to about 1.0 in 0.1% trifluoroacetic acid (TFA) dichloromethane (DCM). The cuvette was irradiated with 660 or 740 nm monochromatic light at various times, and absorption spectra were measured immediately. The slopes of absorbance of DPBF at 415 nm versus irradiation time were calculated and used to compare the $^1$O$_2$ generation ability.

$^1$O$_2$ quantum yield determination of **Lyso700D** (ps) using Methylene Blue (MB) in DCM as reference. ΦΔ was calculated by the following equation:

$$\Phi_{\Delta(ps)} = \Phi_{(MB)} \left( \frac{A_{ps}}{A_{MB}} \right) \left( \frac{F_{MB}}{F_{ps}} \right) \left( \frac{PF_{MB}}{PF_{ps}} \right) \tag{1}$$

Where $A$ is the slope of a plot of the change in absorption of DPBF at 415 nm, $F$ is the absorption correction factor, which can be determined by $F = 1 - 10^{-OD}$, and PF is an absorbed photonic flux (μ Einstein dm$^{-3}$ s$^{-1}$). $\Phi_{MB}$ (0.57) is the $^1$O$_2$ quantum yield of MB in DCM.

### Fluorescence lifetime measurements
Lyso700D (620 nM) in DCM was placed in a long-neck closed cuvette. Nanosecond fluorescence lifetime measurements were performed by using the HORIBA Jobin Yvon IBN photon counting fluorescence system with nano-light emitting diode (LED) excitation at 462 nm. Fluorescence emission lifetime decay curves were analyzed and fitted to obtain the dye molecule's fluorescence lifetime value according to the lifetime decay curve.

### Nanosecond time-resolved transient absorption spectra
The triplet lifetimes of **Lyso700D** were recorded on an LP920 laser flash photolysis spectrometer (Edinburgh Instruments Ltd.) in combination with an Nd:YAG laser (Surelite I-10, Continuum Electro-Optics, Inc.). **Lyso700D** (1.25 μM) in deaerated DCM were excited by a 532 nm laser pulse (1 Hz, 100 mJ per pulse, fwhm ≈7 ns) at room temperature. The triplet state decay kinetics was measured at 684 nm.

### Cell and culture conditions
Raw264.7 was obtained from the Institute of Basic Sciences (IBMS) of the Chinese Academy of Medical Sciences (CAMS). The cell clones were cultured in DMEM supplemented with 10% fetal calf serum (FBS) and streptomycin (0.1 mg ml$^{-1}$) at 37 °C in 95% air with 5% CO$_2$. When used for imaging, all types of cells were cultured on 35 mm glass-bottom culture dishes for 12–24 h.

### Fluorescence Imaging
Raw264.7 cells in the exponential phase of growth were grown on 35 mm glass-bottom culture dishes with 1 mL 90% Dulbecco's modified Eagle Medium (DMEM) + 10% fetal bovine serum (FBS) for 24–36 h to reach 80% confluency in an atmosphere of 95% air with 5% CO2 for 3 h at 37 °C. Then 1 mL fresh medium containing **Lyso700D** (250 nM) was added to incubate for another 3 h. The culture medium was removed and the cells were washed three times with PBS. According to different experimental purposes, the cells were further stained by DND-26 (1 μM), Rhodamine 123 (1 μM), or Hoechst 33342 (2 mg/mL). Fluorescence imaging measurements were performed on confocal laser scanning microscopy (Olympus Fluoview FV1000). The excitation wavelength for **Lyso700D** was 635 nm, while the excitation wavelength for DND-26 and Rhodamine 123 was 488 nm, and for Hoechst 33342 was 405 nm. The detection wavelength was collected from 655 to 755 nm for **Lyso700D**, 500 to 540 nm for DND-26 and Rhodamine 123, and 440 to 480 nm for Hoechst 33342.

### The standard method to prepare D-RAWs
**For fluorescence imaging.** Raw264.7 cells in the exponential phase of growth were grown on 35 mm glass-bottom culture dishes at $5 \times 10^4$ cells and for 24 h in an atmosphere of 95% air with 5% CO$_2$ for 3 h at 37 °C. Then the cells were stained with 250 nM **Lyso700D** in fresh culture medium for another 3 h. The culture medium containing **Lyso700D** was removed and the cells were washed three times with PBS. D-RAWs were obtained by trypsin digestion and centrifugation collection. The cells were dispersed in ethanol and placed for 20 min at room temperature. The above mixture was centrifuged at 800g for 5 min to collect the supernatant was collected. The Lyso700D amount of dye taken by cells was calculated as follows: Firstly, the fluorescence intensity of **Lyso700D** in the supernatant was measured on a fluorescence spectrophotometer (SI Fig. 2a). Subsequently, the fluorescence intensity was converted to the corresponding concentration using a standard curve of the fluorescence intensity versus the concentration of **Lyso700D** in ethanol (SI Fig. 2c). Lastly, the average

amount of **Lyso700D** in each cell ($n_{single\ cell}$) was calculated as $2.56 \pm 0.03 \times 10^{-15}$ mole (SI Table 1).

**For mouse experiments.** Raw264.7 cells in the exponential phase of growth were grown on a T25 cell culture flask at $2 \times 10^5$ cells per flask for 24 h. By the same method as above (SI Fig. 2b), we were able to obtain the average amount of **Lyso700D** of $2.24 \pm 0.07 \times 10^{-15}$ mole in each cell (SI Table 1). For the epidermis model experiment, $2 \times 10^6$ D-RAWs in PBS were injected intravenously in each mouse infected with leg epidermis with an equivalent dose of 0.32 mg/kg. For the meningitis model experiment, $1 \times 10^5$ D-RAWs were injected intravenously into each meningitis mouse with an equivalent dose of 0.016 mg/kg.

The average amount of **Lyso700D** in each cell:

$$n_{single\ cell} = \frac{c \times V}{N} \qquad (2)$$

$n_{single\ cell}$ is the average amount of Lyso700D in each cell, $c$ is the concentration of **Lyso700D** in ethanol; $V$ is the volume of ethanol solution in the test; $N$ is cell numbers in the test.

Equivalent dose:

$$C = \frac{n_{single\ cell} \times N \times M}{m_{mouse}} \qquad (3)$$

$$C = \frac{c \times V \times M}{m_{mouse}} \qquad (4)$$

$n_{single\ cell}$ is the mole of **Lyso700D** in each RAW264.7 cell; $N$ is cell numbers; $M$ is molecular weight of **Lyso700D**; $m_{mouse}$ is the weight of mice.

The equivalent dose of photosensitizer was reported in vivo in the literature.

$$C = \frac{c \times V \times M}{m_{mouse}} \qquad (5)$$

$c$ is the concentration of photosensitizer administered; $V$ is the photosensitizer delivery volume; $M$ is the molecular weight of photosensitizer; $m_{mouse}$ is the weight of the mouse.

### Intracellular Lyso700D retention assay
Raw264.7 cells in the exponential phase of growth were grown on the 6-well plate at $5 \times 10^4$ cells per well for 24 h in an atmosphere of 95% air with 5% $CO_2$ for 3 h at 37 °C. Then the culture medium was removed and fresh culture medium with 250 nM **Lyso700D** was added to incubate for another 3 h. The medium was removed and the cells were washed three times with PBS. Fresh medium was added to the dish again. After 0, 24, 48, and 72 h incubation respectively, cellular uptake **Lyso700D** was quantified by the confocal laser scanning microscopy (Olympus Fluoview FV1000) and flow cytometer (LSRII, BD), respectively.

### Safety evaluation of D-RAWs antimicrobial strategy in simulated normal tissues
Raw264.7 cells were seeded in 35 mm glass-bottom culture dishes and cultured for 24 h. The cell was incubated with 250 nM **Lyso700D** for 3 h in an atmosphere of 95% air with 5% $CO_2$ at 37 °C. Then the medium was removed and the cells were washed three times with PBS. Cos-7 cells were seeded in these dishes and were further incubated for 24 h. After irradiating with 740 nm red LED light (30 mW/cm², 27 J/cm²) for 15 min, the cells were stained with Calcein (AM)/propidium iodide (PI) apoptosis detection kit according to the manufacturer's instructions. The cell apoptosis was visualized by fluorescence microscopy, Calcein

(AM): excitation wavelength of 488 nm, detection wavelength of 515–555 nm; propidium iodide (PI) excitation wavelength of 559 nm, detection wavelength of 590–630 nm.

### Intracellular ROS Detection
Raw264.7 cells were seeded in 35 mm glass-bottom culture dishes and cultured for 24 h. The cell was incubated with 250 nM **Lyso700D** for 3 h in an atmosphere of 95% air with 5% $CO_2$ at 37 °C. Then stained with 1 µM DCFH-DA for another 0.5 h. The medium was replaced by fresh medium and was irradiated with 740 nm red LED light for 15 min (30 mW/cm², 27 J/cm²). Intracellular ROS detection was performed by fluorescence imaging with an excitation wavelength of 488 nm and detection wavelength of 515–555 nm.

### Intracellular phototoxicity by the fluorescent live/dead cell assay
Raw264.7 cells were seeded in 35 mm glass-bottom culture dishes and cultured for 24 h. The cell was incubated with 250 nM **Lyso700D** for 3 h in an atmosphere of 95% air with 5% $CO_2$ at 37 °C. The medium was replaced by fresh medium and was irradiated with 740 nm red LED light for 15 min (30 mW/cm², 27 J/cm²), the cells were stained with Calcein (AM)/propidium iodide (PI) Apoptosis Detection Kit according to the manufacturer's instructions. The cell apoptosis was visualized by fluorescence microscopy, Calcein (AM): excitation wavelength of 488 nm, detection wavelength of 515–555 nm; propidium iodide (PI) excitation wavelength of 559 nm, detection wavelength of 590–630 nm.

### MTT assay
**Concentration-dependent cell viability**. Raw264.7 cells were seeded on 96 well plates ($3 \times 10^3$ cells per well) and cultured for 24 h. **Lyso700D** at different concentrations was added to incubate for 3 h. Then the cells were washed three times with PBS and added fresh medium. The plates were irradiated with 740 nm light for 15 min (30 mW/cm², 27 J/cm²). After irradiation, the cells were incubated for another 24 h. The relative cell viability was measured by MTT assay.

**Cell tissue deep simulation**. Chicken slices were used as cover, to evaluate the photodynamic effect through various depths of tissue. The origin of the chicken tissues is ex vivo frozen chicken breast tissue purchased from Walmart supermarket (used as an experimental consumable).

The cells were seeded on 96 well-plated and were stained with Lyso700D as same as above. Chicken tissues of different thicknesses were covered on the 96-well plates separately before light, and then the MTT assay was operated as same as the above.

### Bacterial culture
Methicillin-resistant *S. aureus* (MRSA, ATCC 43300) and *A. baumannii* (AB, ATCC 19606) were taken from American-type cultures (ATCC, Manassas, VA). The bacteria were cultured overnight at 37 °C with 20 mL Lysogeny Broth (LB) medium (Tryptone 10 g/L, Yeast extract 5 g/L, NaCl 10 g/L). When the bacteria reached a density of the logarithmic growth phase, the bacteria were rinsed with PBS three times and stained with Mem-SQAC (1 µM) for 30 min. After this, the medium was removed and bacteria were rinsed again with PBS three times, and finally, DMEM medium was added to dilute bacterial concentration to achieve the multiplicity of infections.

### Fluorescence imaging of bacteria in D-RAWs
The bacteria prepared above were suspended in PBS and stained with Mem-SQAC (1 µM). The bacteria were incubated with end-to-end rotation for 30 min at room temperature in the dark. After incubation, the unbound dye was removed by washing it 3 times with PBS and resuspended in 1 ml DMEM.

Raw264.7 cells were seeded in 35 mm confocal dishes and incubated with 250 nM **Lyso700D** for 3 h. The cells were washed 3 times with PBS. Then 1 mL DMEM containing Mem-SQAC stained bacteria was added and the cells and bacteria were incubated in an atmosphere of 95% air and 5% $CO_2$ at 37 °C for 30 min. Excess extracellular bacteria were treated with gentamicin sulfate (Sigma-Aldrich, G1914) on the medium for 1 h. The cells were washed 3 times with PBS and observed under a confocal laser scan microscope (Olympus FV1000).

### Light-induced antibacterial assay within D-RAWs

The bacteria prepared above were suspended in PBS and stained with Mem-SQAC (1 μM). Bacteria were incubated with end-to-end rotation for 30 min at room temperature in the dark. After incubation, the unbound dye was removed by washing 3 times with PBS and resuspended in 1 ml DMEM.

Raw264.7 cells were seeded in 35 mm glass-bottom cell dishes and incubated with 250 nM **Lyso700D** for 3 h. The cells were washed 3 times with PBS. Then 1 mL DMEM containing Mem-SQAC stained bacteria was added, and the cells were incubated in an atmosphere of 95% air and 5% $CO_2$ at 37 °C for 30 min. Excess extracellular bacteria were treated with gentamicin sulfate on the medium for 1 h. The cells were washed 3 times with PBS. The cells were illuminated under 740 nm light at different times. and observed under a confocal laser scan microscope (Olympus FV1000).

### Survival of bacteria within D-RAWs post phototherapy

The bacteria were diluted and added to Raw264.7 thin monolayer culture. After co-incubation for 30 min, the cells were treated with 100 μg/mL gentamicin sulfate (Sigma-Aldrich, G1914) for 1 h to kill the extracellular bacteria. After incubation, the cells were washed with PBS and further incubated in 10% FCS-DMEM. The cells were washed with PBS to remove extracellular bacteria and dead cells. Finally, 0.5% Triton X-100 (Sigma-Aldrich, X100) to lyse and a series of dilutions of the solution was seed on TSB agar (Sigma-Aldrich, 22092 and BD Biosciences, 214010) for bacteria counting.

### Animals

All animal experimental models were constructed to be either immunodeficient or immunocompetent, so gender was not considered in the study design and analysis. Feeding conditions for all animals: dark rearing cycle 14 h, light rearing cycle 10 h; ambient temperature 24–29 °C; relative humidity 40–70%.

### Construction of immunodeficiency model in mice and rats

The mouse immunodeficiency models constructed by intraperitoneal administration of cyclophosphamide have been widely used and studied[3–5]. Cyclophosphamide works through the alkylation of DNA, which causes damage to the genetic material of rapidly dividing immune cells (B cells and T cells) and prevents them from effectively dividing and proliferating. Additionally, cyclophosphamide can also induce a shift in the balance of immune cells toward more suppressive immune cells, such as regulatory T cells and B cells, which further inhibits the overall immune response.

We monitored and reported the following items in the revised manuscript: the number of WBCs, LYMs, MONs, and GRANs in the blood during the three consecutive days of CY administration (SI Fig. 8). In addition, we also monitored the number of WBCs, LYMs, MONs, and GRANs in the blood of mice 24 h after bacterial infection (SI Fig. 8). We found a continuous decrease in the number of WBCs and LYMs upon the administration of CY, and a sharp decrease in the number of MONs and GRANs on the third day of CY administration (SI Fig. 8). Lastly, the number of WBCs, LYMs, MONs, and GRANs in the blood of CY-treated mice in the PBS group remained low after bacterial infection, indicating the immune system of these mice collapsed and was unable to respond to infection.

### Treatment of MRSA-induced epidermal inflammation in immunodeficient mice

Female C57BL/6 mice, 6–7 weeks old, were purchased from Dalian Medical University, Shanghai Experimental Animal Center. The animal experiments conducted in this work were approved by the Animal Care and Use Committee of Dalian Medical University. The mouse immunodeficiency models and healthy mice were constructed separately based on literature[3–5]. Briefly, C57BL/6 mice were injected intraperitoneally with 100 mg/kg CY for bacterial infection for 3 consecutive days. Blood routine monitoring of mouse WBCs and LYMs was performed to assess the development of immune deficiency and health recovery in mice.

For the construction of the mouse leg epidermal infection model: the mice were anesthetized with 10% chloral hydrate in PBS intraperitoneally. Then, the epidermis on the thigh of the mouse was cut off to construct a wound surface. 50 μL MRSA suspension ($1 × 10^8$ CFU) was inoculated on the wound surface and covered with a sterile inorganic cloth.

For dynamic tracking D-RAWs in vivo: $2 × 10^6$ D-RAWs were injected intravenously into each mouse, and the body fluorescence images were recorded using a small animal fluorescence imaging system (NightOWL II LB983) at 12, 24, 48, and 72 h, respectively, and the average fluorescence intensity of the inflammatory area was counted using the NightOWL II LB983 system.

For treatment procedure and data collection: At 24 h after MRSA injection, $2 × 10^6$ D-RAWs were injected intravenously into each mouse. The wounds of mice were irradiated with a 740 nm LED light (50 mW/cm², 90 J/cm²) for 30 min/time at 36, 42, or 50 h after the bacteria injection, respectively. The mice's facial veins, wounds, leg, and main organs (heart, liver, spleen, lungs, and kidneys) were collected. Bacterial CFU was quantified at 36 h and 50 h after the bacteria injection. The survival and body weight of mice were assessed every 24 h. After 30 days, the mice were sacrificed. The wound tissue of the mice was aseptically homogenized and bacterial CFU was quantitatively determined.

### Treatment of MRSA-induced epidermal inflammation in immunocompetent mice

**Female C57BL/6 mice, 6–7 weeks old, normal immune status is normal.** For the construction of the mouse leg epidermal infection model: the mice were anesthetized with 10% chloral hydrate in PBS intraperitoneally. Then, the epidermis on the thigh of the mouse was cut off to construct a wound surface. 50 μL MRSA suspension ($1 × 10^8$ CFU) was inoculated on the wound surface and covered with a sterile inorganic cloth.

For dynamic tracking D-RAWs in vivo: $2 × 10^6$ D-RAWs were injected intravenously into each mouse, and the body fluorescence images were recorded using a small animal fluorescence imaging system (NightOWL II LB983) at 12 h and the average fluorescence intensity of the inflammatory area was counted using the NightOWL II LB983 system.

For treatment procedure and data collection: At 24 h after MRSA injection, $2 × 10^6$ D-RAWs were injected intravenously into each mouse. The wounds of mice were irradiated with a 740 nm LED light (50 mW/cm², 90 J/cm²) for 30 min/time at 36, 42, or 50 h after the bacteria injection, respectively. The mice's facial veins, wounds, leg, and main organs (heart, liver, spleen, lungs, and kidneys) were collected and bacterial CFU was quantified at 50 h after the bacteria injection. The survival and body weight of mice was assessed every 48 h. After 30 days, the mice were sacrificed. The wound tissue of the mice was aseptically homogenized, and bacterial CFU was quantitatively determined. Blood routine monitoring of mouse WBCs and LYMs was performed to assess the development of the mice's immune status and health recovery. The main organs were stained with H&E to evaluate treatment outcomes.

## Treatment of MRSA/AB-induced meningitis in immunodeficient mice

**The operation steps for constructing a mouse immunodeficiency model are the same as those described above.** For the construction of the mouse meningitis infection model: The mice were anesthetized with 10% chloral hydrate in PBS intraperitoneally. 10 μL MRSA (20 CFU) or AB (100 CFU) suspension was injected into the CSF layer by the assisted stereotactic apparatus and the wound was sutured.

For D-RAWs crossing the BBB in vivo: Four groups of experiments were set up respectively: (1) blank-- D-RAWs cultured in vitro; (2) control-- healthy mice CSF; (3) D-RAWs-- immunodeficient mice CSF treated with D-RAWs intravenously; (4) D-RAWs+ bacteria-- immunodeficient mice with meningitis CSF treated with D-RAWs intravenously. At 12 h after the administration of D-RAWs, the mouse CSF was centrally extracted and diluted 10-fold with PBS. Fluorescence imaging and flow cytometry analysis were performed.

For treatment process and data collection: At 2 h after MRSA/AB injection, $1 \times 10^5$ D-RAWs were injected intravenously into each mouse. The intact brain bone of mice cranial were irradiated with a 740 nm LED light (50 mW/cm$^2$, 90 J/cm$^2$) for 30 min/time at 14, 20, or 28 h after the bacteria injection respectively. The mice's facial veins and brains were collected and the bacterial CFU was quantified at 28 h after the bacteria injection. The survival and body weight of mice were assessed every 24 h. After 30 days, the mice were sacrificed. The main organs (brain, heart, liver, spleen, lungs, and kidneys) were aseptically homogenized and bacterial CFU was quantitatively determined. Blood routine monitoring of mouse WBCs and LYMs was performed to assess the development of the mice's immune status and health recovery. Brains were stained with H&E to evaluate treatment outcomes.

## Treatment of MRSA-induced meningitis in immunodeficient rats

Female 7–8 weeks old SD rats, were purchased from Dalian Medical University, Shanghai Experimental Animal Center. The animal experiments conducted in this work were approved by the Animal Care and Use Committee of Dalian Medical University. The operation steps for constructing a rat immunodeficiency model are the same as those described above.

**For the construction of the mouse meningitis infection model.** Rats were anesthetized with 25% chloral hydrate in PBS intraperitoneally. 10 μL MRSA (200 CFU) suspension was injected into the CSF layer by the assisted stereotactic apparatus and the wound was sutured.

For isolation of rats' bone marrow cells: the femur and tibia bones were isolated from rats the hair was rinsed off by PBS, and then the bone was cut open. A 21 G needle and 10 mL syringe filled with cold PBS + 2% heat-inactivated FBS were chosen to flush out marrow (3–5 mL/mouse). The marrow through a 21 G needle 4–6 times to dissociate the cells into PBS. Then the cells dispersed in PBS were passed cells through a 70 μm cell strainer to remove cell clumps, bone, hair, and other cells/tissues. Three volumes of NH$_4$Cl solution (0.8% NH$_4$Cl solution, Stemcell Technology) were added and the cells were incubated on ice for 10 min to remove red blood cells. The cell solution was centrifuged at 500×*g* for 5 min at 4 °C. The cell pellet was resuspended in cold PBS + 2%FBS (20–50 ml, depending on the number of cells).

**For BMDM growth medium.** The obtained single cells were dispersed in a medium (DMEM + 10% FBS + 15% filtered + 10 ng/mL M-CSF (monocyte-colony stimulating factor)). The medium was exchanged with fresh BMDM growth medium on day 3.

**For the treatment process and data collection.** At 2 h after MRSA injection, $1 \times 10^6$ D-BMDMs were injected intravenously into each rat. We used implanted glass fiber to conduct light irradiation through the lumbar cistern (671 nm, 30 min, 80 mW/cm$^2$, 144 J/cm$^2$) at 14 h after the bacteria injection. To ensure the consistency of the experimental conditions, the rats in control groups were also treated with the same phototherapy through implanted optical fibers. The rats' facial veins, wounds, and the main organs (heart, liver, spleen, lungs, and kidneys) were collected, and the bacterial CFU was quantified at 26 h after the bacteria injection. The survival and body weight of the rats were assessed every 48 h. After 30 days, the rats were sacrificed and the brain and main organs (brain, heart, liver, spleen, lungs, and kidneys) were aseptically homogenized, and bacterial CFU was quantitatively determined. Blood routine monitoring of rat WBCs and LYMs was performed to assess the development of the rats' immune status and health recovery.

## Statistics and reproducibility

All experiments were performed at least three times ($n \geq 3$) for each sample. All data are presented as the mean ± standard deviation. All data is reproducible. Comparisons between two groups were performed using an unpaired, two-tailed *t*-test. *$P$ < 0.05, **$P$ < 0.01 and ***$P$ < 0.001 were considered significant. The fluorescence images statistical analysis was performed with Olympus FV1000 software. Statistical analysis and graphs were generated using GraphPad Prism and Origin 2018 software. The experiments were randomized. Animal experiments were not considered for sex analysis.

## Reporting summary

Further information on research design is available in the Nature Portfolio Reporting Summary linked to this article.

## Data availability

All data generated or analyzed during this study are included in this published article (and its supplementary information files). Source data are provided in this paper. Source data are provided in this paper.

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

## Acknowledgements

This work was supported by the National Natural Science Foundation of China (Nos. 22278059, 22174009, 22078047, 21901031, and 81902546), Fundamental Research Funds for the Central Universities (Nos. DUT22LAB601, DUT22LAB608, DUT21YG131, and DUT21YG126), Science and Technology Foundation of Liaoning Province (2020-YQ-08), Liao Ning Revitalization Talents Program (No. XLYC1807255), Shenyang Municipal Science and Technology Bureau Medical-Industrial Joint Project (No. 213958), Dalian Science and Technology Innovation Fund (No. 2020JJ25CY014).

## Author contributions

Z.W., A.W., W.C., and Y.L. contributed equally to this work. Conceptualization of the study was done by X.Z, W.C., and Y.X.; the experimental design was carried out by Z.W., X.Z., W.C., and Y.X.; chemical synthesis was carried out by Z.W. and L.W.; cell experiments were carried out by Z.W and D.L.; the mouse model was constructed by Z.W., Y.L., A.W. and W.C.; analysis of results was done by X.Z., Z.W., W.C., and Y.X.; the paper was written by X.Z., Z.W., W.C., and Y.X.; the project was conceived and supervised by X.Z., W.C., and Y.X.

## Competing interests

The authors declare no competing interests.
