## [Peer Review File · Nature Communications]

Adoptive macrophage directed photodynamic therapy of multidrug-resistant bacterial infectionEditorial Note: This manuscript has been previously reviewed at another journal that is not operating a transparent peer review scheme. This document only contains reviewer comments and rebuttal letters for versions considered at *Nature Communications*.

REVIEWER COMMENTS

Reviewer #1 (Remarks to the Author):

The authors have addressed all of my comments. They have included new data and have revised the discussion as requested.

Despite using assistance in English language usage the paper requires more revision from someone with editorial expertise. I will indicate the changes that I recommend for the title and abstract

1. Title line 3. Hosts rather than Host
2. Page 1 Line 12. Clinical infections
3. Page 1 Line 13. Immunodeficiencies
4. Page 1 Line 14. Antibiotic
5. Page 1 Line 20 delete ability
6. Page 1 Line 23. Insert the before photosensitizer
7. Page 1 Line 24. Photodynamic, lower case P
8. Page 2, line 3. At 100%
9. Page 2 Line 7. A proven clinical technique
10. Page 2 Line 8. 100% of infected rats survived
11. Page 2 Line 9. While none of the control group survived
12. Page 2 Line 11. Infections

The introduction needs sharpening to make it more succinct. Here is a suggested rewrite of the first paragraph

Multidrug resistant bacteria are a leading cause of infection. They infect more than 25% of patients in Intensive Care Units (ICU) where they cause a high level of mortality. Strong prophylactic procedures and a high level of clinical care are required in modern clinical medicine. Nevertheless some infections are inevitable. The evolution of MDR bacteria combined with more invasive surgical procedures involving implanted devices has resulted in more severe invasive infections. This is particularly so for immunodeficient patients such as those undergoing chemotherapy. Immunodeficient patients have a high risk of developing severe bacterial infections leading to life threatening sepsis. These patients cannot mount an effective immune response due to failure to produce antibodies, diminished phagocytic activity and impaired T cell function. The ability to treat infections with antibiotics is sometimes ineffective due to antibiotic resistance and the inaccessibility of some sites of infection such as the meninges where the blood brain barrier hinders drug access. This makes treatment of bacterial meningitis with antibiotics particularly difficult. In addition administration of high doses of antibiotic over a prolonged time period leads to serious side effects in immunodeficient patients. Novel effective and broad spectrum therapies as an alternative to antibiotics are sorely needed.

Reviewer #3 (Remarks to the Author):

The authors have addressed the great majority of the points raised by the three reviewers, including in vitro and in vivo new data. I believe this present version of the manuscript may be accepted for publication at Nat Comm if some minor revisions are performed.

In general, the manuscript has been improved considering the language. Even though there are still some typo and grammatical errors, especially at the Supplementary Info document.

In all figures presenting the CFU graphs (figures 3, 4, and 5 in the main manuscript, but also the ones presented in SI), must be presented in log scale. Do the authors performed the bacterial counting in other dilutions? For an antimicrobial effect, only the results of inactivation over 3 log (>99.9%) are relevant.

The authors gave more detailed information concerning the irradiation procedure at the animal models, but still is lacking the interval between each of the 3 irradiation sessions of 30 min. (p.13 lines 19-21; p20 line 1; and SI).

The photosensitizer concentrations between 1.13 and 75 mg/kg BW are extremely high. The authors should double check this information, as well as include other studies that use much lower concentration levels, especially in topical photosensitizer application at the infected wounds.

Reviewer #4 (Remarks to the Author):

The authors have largely responded well to the previous reviews, and the experiments in immunocompetent mice are greatly appreciated. However, additional attention is required to points made in the prior criticism:

1) English grammar and spelling still need significant revision throughout the paper. Specific examples pointed out in prior reviews were not addressed (e.g. Reviewer 3, question 1).

2) Reviewer 2, Major comment 1: Further explanation related to immunodeficient hosts would be helpful. Figure 4 shows that administration of RAWs to immunodeficient mice resulted in improved survival and reduced CFU in the blood. So the observed efficacy of D+RAWs in this case is at least partially due to the administration of macrophages.

3) Reviewer 2, Major comment 2: Can the authors further justify the statement that "fluorescence resulted from the attraction of bacteria in the organs" in response to this comment? Is it possible that fluorescence throughout multiple organ systems is due to off-target uptake? This would be of concern for the specificity of the treatment.

4) Reviewer 2, Major comment 3: Are there concerns for application to biofilms or other localized infections? This therapeutic approach seems to rely on circulation of pathogens after phagocytosis, which is not necessarily a fair assumption. This is particularly true for focal (e.g. abscess) infection.

5) Reviewer 2, Major comment 3: Additionally, this therapeutic approach relies on localized light delivery to a dynamic population of circulating cells. In a larger animal or human, would the fluence deposited during this transit be sufficient to induce a photodynamic effect?

6) Reviewer 2, Major comment 4: Further detail is required on the time necessary for generation of the Lyso700D+macrophage construct, in order to motivate future clinical translation. "Within a couple of hours" is not rigorous, and this timing is crucial for intervention.

7) Reviewer 2, Major comment 5: Are there concerns that the time of treatment is relatively close to the time at which the infection was lethal in untreated mice? Did the authors examine shorter drug-light intervals?

8) Reviewer 3, Line 270 comment: Fluence values in J/cm² should be added, in addition to treatment time, as this is a standard PDT parameter. Additionally, "irradiance" or "fluence rate" are more generally used to describe the optical power per unit area.

Additional comments on revised manuscript:

9) p. 3, lines 17-19: The effects of conventional antimicrobial PDT on healthy tissue are overstated. Would recommend softening this sentence further to reflect the current clinical literature.

10) p. 8, lines 2-3: Methylene blue is not FDA approved as a photosensitizer. It is approved as a treatment for methemoglobinemia.

11) Figure 3: Did the authors look at the effects of Lyso700D alone in cells? How do the results in Figure 3 compare to the free photosensitizer?

12) P. 12, section 5: Can the authors better justify these experiments? If the site of infection is known, why inject the photosensitizer systemically? If the infection site is unknown, how is the site of illumination determined?

13) p. 13, lines 19-21: What is the rationale for multiple treatments? Do the authors have data on the effects of a single illumination session?

14) Figure 4: Did the authors look at times shorter than 12 hours post-injection? Particularly for the Lyso700D only group, uptake in the wound may peak at a time earlier than 12 hours. So light may be delivered at a sub-optimal time for this group, biasing the results towards D+RAWs.

15) Figure 4c: Would recommend using a more standardized log scale for the y axes, as the scales used make interpretation more difficult.

16) P. 22, lines 27-29: What irradiance was used for these experiments?

17) Figure 7c: Did control animals also receive light? This is not obvious in the text.

Dear editor and referees,

We thank the referees for the comments and suggestions to improve our manuscript and refresh our understanding of this work. I've read these comments and suggestions carefully and revised our manuscript accordingly. We have further improved the English writing and added two more in vivo experiments to address and respond to all the concerns. Below is the point-by-point reply to the comments and suggestions.

Reviewer #1:

The authors have addressed all of my comments. They have included new data and have revised the discussion as requested.

Despite using assistance in English language usage the paper requires more revision from someone with editorial expertise. I will indicate the changes that I recommend for the title and abstract

1. Title line 3. Hosts rather than Host
2. Page 1 Line 12. Clinical infections
3. Page 1 Line 13. Immunodeficiencies
4. Page 1 Line 14. Antibiotic
5. Page 1 Line 20 delete ability
6. Page 1 Line 23. Insert the before photosensitizer
7. Page 1 Line 24. Photodynamic, lower case P
8. Page 2, line 3. At 100%
9. Page 2 Line 7. A proven clinical technique
10. Page 2 Line 8. 100% of infected rats survived
11. Page 2 Line 9. While none of the control group survived
12. Page 2 Line 11. Infections

The introduction needs sharpening to make it more succinct. Here is a suggested rewrite of the first paragraph

Multidrug resistant bacteria are a leading cause of infection. They infect more than 25% of patients in Intensive Care Units (ICU) where they cause a high level of mortality. Strong prophylactic procedures and a high level of clinical care are required in modern clinical medicine. Nevertheless some infections are inevitable. The evolution of MDR bacteria combined with more invasive surgical procedures involving implanted medical devices has resulted in more severe invasive infections. This is particularly so for immunodeficient patients such as those undergoing chemotherapy. Immunodeficient patients have a high risk of developing severe bacterial infections

leading to life threatening sepsis. These patients cannot mount an effective immune response due to failure to produce antibodies, diminished phagocytic activity and impaired T cell function. The ability to treat infections with antibiotics is sometimes ineffective due to antibiotic resistance and the inaccessibility of some sites of infection such as the meninges where the blood brain barrier hinders drug access. This makes treatment of bacterial meningitis with antibiotics particularly difficult. In addition administration of high doses of antibiotic over a prolonged time period leads to serious side effects in immunodeficient patients. Novel effective and broad spectrum therapies as an alternative to antibiotics are sorely needed.

Thanks for your comments and kind suggestions. We have further improved the English language in the revised manuscript and supporting information. We also revised the introduction according to your suggestion. Your suggestions improve the quality of our manuscript.

Reviewer #3

The authors have addressed the great majority of the points raised by the three reviewers, including in vitro and in vivo new data. I believe this present version of the manuscript may be accepted for publication at Nat Comm if some minor revisions are performed.

In general, the manuscript has been improved considering the language. Even though there are still some typo and grammatical errors, especially at the Supplementary Info document.

Thanks for your comments and suggestions. We have worked with skilled English writers to further improve the writing in the revised manuscript and supporting information and corrected these typos and grammatical errors.

In all figures presenting the CFU graphs (figures 3, 4, and 5 in the main manuscript, but also the ones presented in SI), must be presented in log scale. Do the authors performed the bacterial counting in other dilutions? For an antimicrobial effect, only the results of inactivation over 3 log (>99.9%) are relevant.

Thanks for your comments and suggestions. We have standardized the y-axis of the bacterial load statistics for all Figures in the manuscript and supporting information using a logarithmic scale. For bacterial enumeration, we first diluted the stock solution of obtained bacteria by 10^1 , 10^2 , 10^3 , 10^4 and 10^5 times. We then use 10 μ l of the stock solution and each dilution for plate coating. The plate counting data is consistent with the dilution pattern. These data were collected and used to calculate the bacteria numbers in 1 mL of the stock solutions. Specifically, no bacterial colonies were observed either in the stock solutions or in the dilutions for the experimental groups.

The authors gave more detailed information concerning the irradiation procedure at the animal models, but still is lacking the interval between each of the 3 irradiation sessions of 30 min. (p.13 lines 19-21; p20 line 1; and SI).

Thanks for your comments and suggestions. We have added detailed information to describe the irradiation procedures in the revised manuscript and supporting information.

We added more description and discussion on page 14, line 4 in the revised manuscript:

“We, therefore, treated the infection site with 90 J/cm² light irradiation (the irradiance is much lower than the safety threshold of 300 mW/cm²) for 30 min/time/day for two successive days (a standard method of irradiance in this study, detailed information is shown in supporting information).”

We added more description and discussion on page 17, line 21 in the revised manuscript:

“We then treated the infection site with light irradiation at an irradiance of 90 J/cm² light irradiation for 30 min/time/day for two successive days (a standard method of irradiance in this study, detailed information is shown in supporting information, Fig. 5a).”

We added more description and discussion on page 20, line 29 in the revised manuscript:

“Secondly, we treated the infected mice in two D-RAWs groups over intact brain bone with light irradiation at an irradiance of 90 J/cm² light irradiation for 30 min/time/day for two successive days (a standard method of irradiance in this study, detailed information is shown in supporting information).”

We added more description and discussion on page 10, line 13 in the revised supporting information:

“The wounds of mice were irradiated with a 740 nm LED light (90 J/cm²) for 30 min/time at 36 h, 42 h, or 50 h after the bacteria injection, respectively.”

We added more description and discussion on page 11, line 1 in the revised supporting information:

“The wounds of mice were irradiated with a 740 nm LED light (90 J/cm²) for 30 min/time at 36 h, 42 h, or 50 h after the bacteria injection, respectively.”

We added more description and discussion on page 11, line 23 in the revised supporting information:

“The intact brain bone of mice cranial were irradiated with a 740 nm LED light (90 J/cm²) for 30 min/time at 14 h, 20 h, or 28 h after the bacteria injection, respectively.”

We added more description and discussion on page 12, line 22 in the revised supporting information:

“We used implanted glass fiber to conduct light irradiation through the lumbar cistern (671 nm, 30 min, 144 J/cm²) at 14 h after the bacteria injection.”

The photosensitizer concentrations between 1.13 and 75 mg/kg BW are extremely high. The authors should double check this information, as well as include other studies that use much lower concentration levels, especially in topical photosensitizer application at the infected wounds.

Thanks for the comment and the suggestion. We carefully investigated the literature within the last three years focusing on the doses of organic photosensitizers for mouse epidermal wound models. According to the reported data, the equivalent dose of photosensitizers (calculated according to the

equation below) falls into the range of 8.53 µg/kg -24 mg/kg mice. We summarized the reported doses and the corresponding references in the revised manuscript and supporting information. The method for the calculation of the dose was described in the revised supporting information.

We updated the reported doses on page 15, line 10 in the revised manuscript:

“Importantly, according to our calculation, the equivalent dose (weight of Lyso700D in each cell × cell numbers/weight of mice) is 0.32 mg/kg mice, which is relatively low doses of photosensitizer in recently reported treatment of epidermal bacterial infection (ranging from 8.53 µg/kg -24 mg/kg mice, SI Table 2)^{37,38,43-47}”

We added the method for the calculation of dose and a table to summarize the reported doses and the corresponding references on page 6, line 11 in the revised supporting information:

“9. Equivalent dose of photosensitizer reported in vivo in the literature.

$$C = \frac{c \times V \times M}{m_{mouse}}$$

c is the concentration of photosensitizer administered; *V* is the photosensitizer delivery volume; *M* is the Molecular weight of photosensitizer; *m_{mouse}* is the weight of mice.

Table 2. A summary of the reported doses and the corresponding references for use in epidermal wound models.”

Model	Epidermal wound	Epidermal wound	Epidermal wound	Epidermal wound	Epidermal wound	Epidermal wound	Epidermal wound
Equivalent dose	3.66 mg/kg	1 mg/kg	4 mg/kg	24 mg/kg	2 mg/kg	1 mg/kg	8.53 µg/kg
ref	37	38	41	42	43	44	45

Reviewer #4

The authors have largely responded well to the previous reviews, and the experiments in immunocompetent mice are greatly appreciated. However, additional attention is required to points made in the prior criticism:

1)English grammar and spelling still need significant revision throughout the paper. Specific examples pointed out in prior reviews were not addressed (e.g. Reviewer 3, question 1).

Thanks for your comments and suggestions. We have worked with skilled English writers to improve the writing in the revised manuscript and supporting information and corrected these typos and grammatical errors carefully.

2) Reviewer 2, Major comment 1: Further explanation related to immunodeficient hosts would be

helpful. Figure 4 shows that administration of RAWs to immunodeficient mice resulted in improved survival and reduced CFU in the blood. So, the observed efficacy of D+RAWs in this case is at least partially due to the administration of macrophages.

Thanks for the comment and the suggestion. Following your suggestion, we have added more description and discussion of the immunosuppressed state in the revised manuscript to fully respond to the comments by reviewer 2.

We added more description and discussion on page 2, line 14 in the revised manuscript:

“These patients are insensitive to recognize and respond to bacterial infections due to a lack of immune system activity, including decreased production of key immune cells, complement system function, and dysregulation or deficiency of inflammatory mediators. This enhanced risk is led by several factors, including reduced production of antibodies, diminished phagocytic activity, and impaired T-cell function.”

We added more description and discussion on page 12, line 5 in the revised manuscript:

“Immunodeficient refers to a state in which the immune system’s functions are impaired, resulting in reduced immune cell activity and a weakened ability to fight off infections. In an immunodeficient state, the number of immune cells, such as macrophages, T cells, and natural killer cells, may be reduced, compromising the body’s defense mechanisms. Additionally, the chemotactic activity, which allows immune cells to migrate toward sites of infection, may be diminished. Moreover, along with the reduced function of the complement system, the ability of immune cells to eliminate bacteria may also be compromised, leaving the body more susceptible to microbial invasions.”

3) Reviewer 2, Major comment 2: Can the authors further justify the statement that “fluorescence resulted from the attraction of bacteria in the organs” in response to this comment? Is it possible that fluorescence throughout multiple organ systems is due to off-target uptake? This would be of concern for the specificity of the treatment.

We appreciate this comment and suggestions. According to our strategy, the photosensitizer is loaded within the lysosomes of macrophages and it travels along with macrophages in vivo. The ability to find and capture bacteria we discussed is actually the functional feature of macrophages. Here, to make an appropriate statement on the fluorescence from the organs, we carefully researched the functional features of macrophages in living bodies. According to the literature, normally, macrophages are widely distributed throughout an organism’s organs and tissues, ensuring immune surveillance and tissue homeostasis (*J. Pathol.* 250, 656–666 (2020)). The liver (Kupffer cells), spleen, lungs (alveolar macrophages), and central nervous system (microglia) all have a resident population of macrophages (*Nat. Rev. Immunol.* 17, 451–460 (2017)). In addition to resident macrophages, circulating monocytes in the bloodstream can be recruited to various tissues and differentiate into macrophages in response to inflammation, infection, or injury. However, during an infection, there is a chemotactic gradient in the deployment of macrophages specifically migrating toward the site of pathogen invasion. They phagocytose and destroy the invading pathogens, present antigens to activate other immune cells, release inflammatory mediators to

recruit additional immune cells and promote tissue repair. Here, only a number of macrophages, other than all macrophages, are deployed to fight against bacteria, since our immune system is conservative (*J. Invest. Dermatol.* **127**, 514-525(2007); *Essays Biochem.* **60**, 275–301 (2016)), which means there are still plenty of macrophages sticking to their original location.

To confirm and demonstrate the distribution feature of macrophages, we performed an additional in vivo experiment using healthy mice. We administrated D-RAWs through i.v. injection into healthy mice and harvested their organs 12 h after the administration. Ex-vivo fluorescence imaging showed intense fluorescence in the livers, spleens, and lungs, which was similar to the distribution of D-RAWs in the infection model. We assume that the D-RAWs first distribute throughout the body, and work as part of the body's immune system. From this aspect, the distribution of D-RAWs in the liver and other organs should be regularly governed by the immune system, other than an off-target distribution. To respond to the reviewer's comment, we rationally speculate that the increase in bacteria in the organs of the immunodeficient bacteria-infected mice in Figure 4 would also increase the recruitment of D-RAWs.

Thus, in our strategy, not all of the D-RAWs were employed by the immune system to capture the bacteria. A rational description should be that a fraction of the D-RAWs travel along the bloodstream to capture the bacteria and the rest of them stay in a standby state in the organs. Here, we changed our expression on the working mechanism of D-RAW by using "tracking and capturing" instead of "targeting". We also added more description on the distribution feature of macrophages in the revised manuscript to clarify the rationale. Here, as the reviewer noticed, there was only a small portion of macrophage "finding" the infection site (capturing bacteria) based on the fluorescence intensity. However, according to our therapeutic results (Fig. 4c), this small portion of D-RAWs that reached the infection site was sufficient to eliminate the bacteria at the infection sites. The injected D-RAWs tracked, captured, and eliminated bacteria at the infection sites and in the blood (Fig. 4c). On the contrary, the epidermal bacterial infection in the PBS-treated mice developed bacteremia 36 h after bacteria injection. As the reviewer suggested, we added more description on the distribution feature of macrophages and added the new ex-vivo fluorescence imaging results in the revised supporting information to fully illustrate the rationale of our strategy.

We added more description on the distribution feature of macrophages on page 13, line 19 in the revised manuscript:

"Additionally, we observed significant fluorescence in the main organs through ex vivo imaging (SI Fig. 10). According to the distribution feature of macrophages in the literature, they widely distribute throughout an organism's organs and tissues, ensuring immune surveillance and tissue homeostasis. While, during an infection, there is a chemotactic gradient in the deployment of macrophages specifically migrating toward the site of pathogen invasion. Here, only a number of macrophages, other than all macrophages, are deployed to fight against bacteria, since our immune system is conservative^{41, 42}, which means there are still plenty of macrophages sticking to their original location."

We added ex-vivo fluorescence imaging of organs from healthy mice 12 h after the administration of D-RAWs on page 16 line 1 in the revised supporting information:

SI Figure 10. Ex-vivo fluorescence imaging of organs from healthy mice 12 h after the administration of D-RAWs.

4) Reviewer 2, Major comment 3: Are there concerns for application to biofilms or other localized infections? This therapeutic approach seems to rely on circulation of pathogens after phagocytosis, which is not necessarily a fair assumption. This is particularly true for focal (e.g. abscess) infection.

Thanks for the comment and the suggestion. According to our results, we infer that our method is capable of eliminating biofilms and focal infections (an easier model to treat). Finding the bacteria is the first step of our treatment, and the chemotactic ability of RAW is smart enough to do this for both infection types. The second step is to illuminate the cells that capture the bacteria. According to the working mechanism of our strategy, bacteria should be taken into the lysosome to approach the photosensitizer. In the model of meningitis in the paper, we use the circulation of the cells in the cerebrospinal fluid, and we are able to kill the bacteria that are dispersed in the cerebrospinal fluid by indirectly irradiating the distal irradiation site. For biofilms or other localized infections, we believe they are more feasible to be cured through phototherapy. As the therapeutic results in our manuscript, D-RAWs are able to track and capture the bacteria in epidermal infection. These bacteria can be removed by phototherapy completely. As biofilm is not our primary therapeutic target in this manuscript, we just performed an in vitro phototherapy to demonstrate its potential to cope with biofilm in response to reviewer 1. We constructed biofilms using GFP-expressing *Staphylococcus aureus* (GFP-SA) and treated them with D-RAWs and phototherapy. Two-color confocal imaging showed that D-RAW (red channel) had penetrated into the biofilm microhabitat and phagocytosed some of the bacteria, as evidenced by overlapping portions of the green biofilm and red D-RAW (Fig. 1a). Crystal violet staining showed structural damage to the GFP-SA biofilm after one phototherapy, resulting in a decrease in biomass of about 30% (Fig. 1b). These results demonstrate that D-RAW can kill bacteria in the biofilm. We speculate that D-RAWs are able to find biofilm in vivo as well as they find bacteria in the epidermal infection. However, as the reviewer concerned, an implanted glass fiber is needed to conduct light irradiation at the infection site. We are working on solving this limitation using chemiluminescence in another project. Hopefully, we will be able to share these results in future works.

Fig. 1 (a) 3D confocal imaging of GFP-SA phagocytosed by D-RAWs in a biofilm formation. (b) Photographs (left) and relative biofilm biomass (right) of MRSA biofilms stained by crystal violet after D-RAWs light treatments. Bar: 100 pixels.

5) Reviewer 2, Major comment 3: Additionally, this therapeutic approach relies on localized light delivery to a dynamic population of circulating cells. In a larger animal or human, would the fluence deposited during this transit be sufficient to induce a photodynamic effect?

We appreciate this comment. We are confident with the efficiency of our therapeutic strategy for future translation to large animals or humans. As the reviewer concerned, we had also worried about the efficiency of our therapy before we performed the experiments. However, the in vivo assay showed promising results with a low power density (epidermal infection mice model 90 J/cm², meningitis mice model 90 J/cm², and meningitis rat model 144 J/cm²). Importantly, the total phototherapy duration for all three models is only 90 min. It's feasible to promote the photodynamic effect simply by increasing the power density and duration. In a current study, we verified the photodynamic efficacy when reducing the irradiation power density but prolonging the duration time, and obtained promising results. Hopefully, we will be able to share these results in future works.

6) Reviewer 2, Major comment 4: Further detail is required on the time necessary for generation of the Lyso700D+macrophage construct, in order to motivate future clinical translation. "Within a couple of hours" is not rigorous, and this timing is crucial for intervention.

Following the reviewer's suggestion, we added the detailed preparation of D-RAWs and D-BMDMs in the support information. Basically, it took 3 h to finish the staining process before the administration of these cells into infected hosts. The standard method to prepare D-RAWs is listed on page 5 line 1 in the revised supporting information.

7) Reviewer 2, Major comment 5: Are there concerns that the time of treatment is relatively close to the time at which the infection was lethal in untreated mice? Did the authors examine shorter drug-light intervals?

Thanks to the reviewer for this important question. As the reviewer concerned, we made full preparation, by collecting literature and performing pre-experiments, to set a suitable therapeutic routine. According to our understanding, the treatment time of this epidermal model was set based on three principles: 1) To successfully establish an epidermal infection. According to the literature, it takes at least 24 hours to establish epidermal inflammation. 2) The treatment should precede the time of death of a mouse model. Based on our experience and the literature, the time of death in the mouse infection model was positively correlated with the initial bacterial dose (*PLOS Pathog.* **14**, e1007112 (2018)). According to our results (Figure 4b), it took about 12 h for D-RAWs to capture the bacteria. 3) It should occur after the cells have effectively captured the bacteria. According to our pre-experiment, with the bacterial burden in our assay, the first death of mice happened 48 h after the injection of bacteria. Therefore, we decided to administrate D-RAWs 24 h after the injection of bacteria and perform the irradiation in another 12 h (that is 36 h after the injection of bacteria) to give enough time for D-RAWs to capture bacteria (12 hours is enough as shown in figure 4b.) We did not try an earlier treatment time for our model. According to the results, the current therapeutic routine is suitable to demonstrate the treatment effect.

8) Reviewer 3, Line 270 comment: Fluence values in J/cm² should be added, in addition to treatment time, as this is a standard PDT parameter. Additionally, “irradiance” or “fluence rate” are more generally used to describe the optical power per unit area.

Following the reviewer’s suggestion, we have corrected the description of optical power per unit area and used “J/cm²” as the unit in the revised manuscript and supporting information.

Additional comments on revised manuscript:

9) p. 3, lines 17-19: The effects of conventional antimicrobial PDT on healthy tissue are overstated. Would recommend softening this sentence further to reflect the current clinical literature.

Thanks for the comment and the suggestion. We have revised the expression in the revised manuscript.

We changed the expression on page 3, line 3 in the revised manuscript:

“However, most photosensitizers represented by organic fluorophores lack specificity in targeting the live bacteria. In most cases, current trials using photosensitizers may need an in-situ administration of a high dose of photosensitizers, such as spray on the wounds, to kill bacteria completely. And, sometimes, the over-dose of photosensitizer may damage the healthy tissue of the host if not properly dealt with.”

10) p. 8, lines 2-3: Methylene blue is not FDA approved as a photosensitizer. It is approved as a treatment for methemoglobinemia.

Thanks for the comment. We corrected the expression in the revised manuscript.

We corrected the discussion on page 7, line 8 in the revised manuscript:

“(MB, a popular photosensitizer used in antimicrobial and anti-tumor therapy.)”

11) Figure 3: Did the authors look at the effects of Lyso700D alone in cells? How do the results in Figure 3 compare to the free photosensitizer?

Thanks for the comments. We investigated the effect of free Lyso700D on the activity of RAWs and the antimicrobial effect on MRSA in the manuscript and supporting information. As shown in SI Figure 4a (Dark group), RAWs were treated with various concentrations (120 nM-1250 nM) of Lyso700D for 3 h. MTT results showed equal viability of RAWs for Lyso700D treated groups and the control group, which indicates that free Lyso700D has negligible dark toxicity on RAWs. We also performed phototherapy on MRSA using free Lyso700D. However, as shown in SI Figure 7, there is no difference in the number of bacteria between the control group and Lyso700D groups (with or without irradiation), which indicates free Lyso700D is not able to kill bacteria due to a lack of targeting specificity toward bacteria. Following the reviewer’s suggestion, we added more description and discussion to specify the effects of free Lyso700D on RAWs and bacteria in the revised manuscript.

We added a discussion on the toxicity of free Lyso700D on RAWs on page 10, line 9 in the revised manuscript:

“We also confirmed that free Lyso700D affords excellent ROS generation ability but negligible dark toxicity toward RAWs through MTT assay and live/dead cell assay (SI Fig. 4a, b, d).”

We added a discussion on the antimicrobial effect of free Lyso700D on MRSA on page 9, line 13 in the revised manuscript:

“To fully demonstrate the advantage of our therapeutic strategy over the free photosensitizer molecule, we also performed phototherapy on MRSA using free Lyso700D. However, as shown in SI Figure 7, there is no difference in the number of bacteria between the control group and Lyso700D groups (with or without irradiation), which indicates free Lyso700D is not able to kill bacteria due to a lack of targeting specificity toward bacteria.”

SI Figure 4. (a) Cell viability of D-RAWs with or without irradiation at 740 nm (27 J/cm^2). (b) Concentration-dependent phototoxicity of Lyso700D under 740 nm (27 J/cm^2). (c) Detection of D-RAWs intracellular ROS generated by 740 nm (27 J/cm^2). (d) Confocal images of D-RAWs containing Calcein-AM-PI with or without 740 nm light irradiation (Calcein-AM was collected at 500 nm-550 nm by excitation at 488 nm. PI was collected at 565 nm-650 nm by excitation at 559 nm.). Bar: 20 μm .

SI Figure 7. Survival of MRSA post phototherapy using freshly prepared D-MRSAs.

12) P. 12, section 5: Can the authors better justify these experiments? If the site of infection is known, why inject the photosensitizer systemically? If the infection site is unknown, how is the site of illumination determined?

Thanks for the pertinent comments and concerns. To respond to your concern about experiments in section 5, we'd like to specify the aim of these experiments and the advantage of our strategy to find and capture each bacterium. First, the major goal of building the epidermal model in section 5 is to demonstrate the ability of D-RAWs (through systemic administration) to find the bacteria. In the

epidermal model, bacteria accumulate in the superficial layer of the skin. If D-RAWs do track and capture bacteria in the skin, it is feasible to demonstrate the accumulation of D-RAWs at the infection site through in vivo fluorescence. As shown in Figure 4b, in vivo fluorescence imaging showed more intense fluorescence in the infection site than in control wounds. Here, as the reviewer noticed, we agree that the epidermal model is neither a necessary model to demonstrate the therapeutic advantage of our strategy nor a serious infection worth a new therapeutic strategy. To avoid such concerns as the reviewer raised, we added more discussion to the revised manuscript specifying the major goal of the experiments in section 5.

Further on, we would like to illustrate here that the advantage of our strategy is to find and capture each bacterium, not just accumulate in the infection site. Such bacterium specificity may significantly suppress the side effects of photodynamic effects on healthy tissue. For example, the sprayed photosensitizer generally distributes on the surface layer of the wound, attaching to both the bacteria and the healthy cells. There is a big chance that the photodynamic effect damages healthy cells as well as killing the bacteria. In our strategy, D-RAWs find and phagocytize bacteria into lysosomes, where Lyso-700D are loaded. The photodynamic effect can be applied to the bacteria, and confined within D-RAWs (as demonstrated in Figure 3f). Therefore, our therapeutic strategy promotes the photodynamic effect to bacteria, while restricting the side effects to healthy tissue. To illustrate this advantage, we added more discussion in the revised manuscript.

We added a discussion on the major goal of building an epidermal model on page 12, line 1 in the revised manuscript:

“The major goal of building an epidermal model is to demonstrate the ability of D-RAWs to find the bacteria. In the epidermal model, bacteria accumulate in the superficial layer of the skin. If D-RAWs track and capture bacteria in the skin, it is feasible to demonstrate the accumulation of D-RAWs at the infection site through in vivo fluorescence.”

We added a discussion on the advantage of our strategy on page 10, line 18 in the revised manuscript:

“In our strategy, D-RAWs find and phagocytize bacteria into lysosomes, where Lyso-700D are loaded. The photodynamic effect can be applied to the bacteria, and confined within D-RAWs (as demonstrated in Figure 3f). Therefore, our therapeutic strategy promotes the photodynamic effect to bacteria, while restricting the side effects to healthy tissue.”

13) p. 13, lines 19-21: What is the rationale for multiple treatments? Do the authors have data on the effects of a single illumination session?

Thanks for the comments. We decided to perform three phototherapies mainly based on the antibacterial effect (through the plate counts of bacteria) post-phototherapy. According to plate counts of bacteria (Figure 4c, 50 h), three times of phototherapy can eliminate bacteria completely. As the reviewer mentioned, we do have the plate counts result after a single phototherapy. As shown in Figure 4c, the image of the plate at 36 h and the result shows the bacterial load of leg wounds and blood from mice after a single phototherapy. The plate statistics of legs show that 94.54% of bacteria were eliminated in the D-RAWs (i.v.) group and 96.57% of bacteria were eliminated in the D-RAWs

(i.v. + i.p.) group post one phototherapy. In order to completely remove the in-situ bacteria and prevent the recurrence of the infection, we decided to perform three times of phototherapy. As shown in Figure 4c, the bacteria were completely eliminated after the third phototherapy. As the reviewer suggested, a detailed evaluation of the dose of illumination, including the fluence rate, illumination times, and duration, should be performed in future studies.

14) Figure 4: Did the authors look at times shorter than 12 hours post-injection? Particularly for the Lyso700D only group, uptake in the wound may peak at a time earlier than 12 hours. So light may be delivered at a sub-optimal time for this group, biasing the results towards D+RAWs.

Thanks for the comment and the suggestion. We didn't perform *in vivo* fluorescence imaging shorter than 12 h in our original manuscript. As the reviewer suggested, we also think it is worth investigating the distribution of D-RAWs and Lyso700D in live mice respectively to perform the phototherapy at a suitable time. We, therefore, performed an additional *in vivo* experiment to evaluate the dynamic distribution of D-RAWs and Lyso700D in mice (the same mice model as that in Figure 4) at 1 h, 3 h, and 6 h respectively. As shown in Figure 9 below, there is almost no fluorescence on the inflammation site of the right leg or the common wound of the left leg at 1, 3, and 6 h in both groups (SI Figs. 9). For the D-RAWs group, this result confirmed that D-RAWs did not migrate to the bacterial/infection site in large amounts within the first 6 h, which indicates the phototherapy window is 6 h after the injection of D-RAWs. For the Lyso700D group, this result confirmed that Lyso700D showed no specificity to target the infection site in the time range of 1-72 h (Fig. 4) and free Lyso700D was not applicable for phototherapy of bacterial infection.

We added a detailed description on page 13 line 12 in the revised manuscript:

“We also evaluated the distribution feature of D-RAWs and Lyso700D in a shorter time. As shown in SI Figure 9, there is almost no fluorescence on the inflammation site of the right leg or the common wound of the left leg at 1, 3, and 6 h in both groups. These results fully confirm the ability of D-RAWs to track bacteria/ infection *in vivo*. Whereas, Lyso700D showed no specificity to target the infection site in the time range of 1-72 h after the administration.”

We added small animal imaging data on page 15 line 8 in the revised supporting information:

Figure 9. (a) Fluorescence images of infected mice at 1, 3, and 6 h post injecting D-RAWs and statistical analysis of the fluorescence intensity of the wounds on legs. (b) Fluorescence images of infected mice at 1, 3, and 6 h post injecting Lyso700D and statistical analysis of the fluorescence intensity of the wounds on legs.

15) Figure 4c: Would recommend using a more standardized log scale for the y axes, as the scales used make interpretation more difficult.

Thanks for the suggestion. We have standardized the y-axis of the bacterial load statistics for all Figures in the manuscript and supporting information using a logarithmic scale.

16) P. 22, lines 27-29: What irradiance was used for these experiments?

Thanks for the suggestion. We have added detailed information in the revised manuscript.

We added a related discussion on page 23 line 29 in the revised manuscript:

“After one phototherapy (30 min, 144 J/cm², LED light with a wavelength of 671 nm), all rats survived and lived beyond the experiment time of 30 days.”

17) Figure 7c: Did control animals also receive light? This is not obvious in the text.

Thanks for the suggestion. The rats in the control group were also treated with the same phototherapy through implanted optical fibers.

We added a detailed description of the operation on page 12 line 23 in the revised supporting information:

“To ensure the consistency of the experimental conditions, the rats in control groups were also treated with the same phototherapy through implanted optical fibers.”

Thank you for your kind consideration.

Sincerely,

Xinfu Zhang

REVIEWERS' COMMENTS

Reviewer #4 (Remarks to the Author):

The authors have largely responded well to the previous reviews, and the additional experiments are greatly appreciated. However, some additional revisions are requested:

- 1) Extensive English language editing is still required throughout the paper. Much of the added text requires revision, including the Methods section.
- 2) Reviewer 4, comment 2: The authors did not respond to the question regarding whether "observed efficacy of D+RAWs in this case is at least partially due to the administration of macrophages." Can the authors highlight the relative importance of macrophages only, compared to D+RAWs?
- 3) Reviewer 4, comment 3: The response added on p. 13 is not written rigorously. This should be revised to improve rigor.
- 4) Reviewer 4, comment 3: Can the authors label the organs shown in SI Figure 10 and SI Figure 11 for clarity?
- 5) Reviewer 4, comment 8: The original review was not asking to remove irradiance in mW/cm², but to additionally include fluence in J/cm². Irradiance values should be re-inserted.

Dear editor and referees,

We thank the referees for the comments and suggestions to improve our manuscript. I've read these comments and suggestions carefully and revised our manuscript accordingly. We've prepared the complete files as required. Below is the point-by-point reply to the comments and suggestions.

Reviewer #4:

The authors have largely responded well to the previous reviews, and the additional experiments are greatly appreciated. However, some additional revisions are requested:

1) Extensive English language editing is still required throughout the paper. Much of the added text requires revision, including the Methods section.

Thanks for your comments and suggestions. We have improved the English language in the revised manuscript thoroughly.

2) Reviewer 4, comment 2: The authors did not respond to the question regarding whether “observed efficacy of D+RAWs in this case is at least partially due to the administration of macrophages.” Can the authors highlight the relative importance of macrophages only, compared to D+RAWs?

We apologize for overlooking this question. According to our understanding and the results we obtained, the supplement of macrophages does help to delay the development of sepsis at the beginning. As shown in Figure 4c, for the PBS-RAWs group, the CFU in the blood system at 36 h is lower than that in the PBS group. This result indicates that the active immune cells in the immunocompetent mice or the supplemented immune cells in the immunodeficiency mice (such as the PBS-RAWs group in our assay) can fight bacterial infections initially to some extent. However, it is worth noting that the antibacterial efficiency is limited, as 50 hours after the infection, the CFU in the blood system for the PBS-RAWs group is almost as high as that in the PBS group. Also, as shown in Figure 4c, the supplemented macrophages showed almost no antibacterial efficiency at the infected leg, indicating only the limited antibacterial efficiency of macrophages. We added more description and discussion in the revised manuscript.

We added more description and discussion on page 14, line 21 in the revised manuscript:

“For the PBS-RAWs group, the treatment delayed the development of sepsis initially as indicated by lower CFU in the blood system (0 CFU at 36 h, $3.2 \pm 0.26 \times 10^7$ CFU at 50 h) compared with PBS group ($2.17 \pm 0.15 \times 10^6$ CFU at 36 h, $7.4 \pm 0.53 \times 10^7$ CFU at 50 h) (Fig. 4c). This result indicates that the active immune cells in the immunocompetent mice or the supplemented immune cells in the immunodeficiency mice (such as the PBS-RAWs group in our assay) can fight bacterial infections initially to some extent; it also reflects the vulnerability of immunodeficient mice against bacterial infections.”

3) Reviewer 4, comment 3: The response added on p. 13 is not written rigorously. This should be revised to improve rigor.

Thanks for your comments and suggestions. We have improved these responses in the revised manuscript.

We added more description on the distribution feature of macrophages on page 13, line 22 in the revised manuscript:

“Additionally, we observed significant fluorescence in the main organs through ex vivo imaging (SI Fig. 10). Considering the longer retention characteristics of D-RAWs than Lyso700D in the organs (SI Fig 11), we can boldly rule out the cause of the leakage of free dye Lyso700D. We infer that fluorescence in the main organs has resulted from two potential reasons. The first one is, according to the distribution feature of macrophages in the literature, they widely distribute throughout an organism’s organs and tissues, ensuring immune surveillance and tissue homeostasis. The second one is the chemotactic gradient drives macrophages specifically to the site of bacterial infection. Here, only a number of macrophages, other than all macrophages, are deployed to fight against bacteria, since our immune system is conservative^{41, 42}, which means there are still plenty of macrophages sticking to their original location. According to the CFU of organ samples (Fig. 4c and SI Fig. 12), the epidermal bacterial infection in the PBS-treated immunosuppressed mice developed into bacteremia 36h after the bacteria injection. D-RAWs, therefore, tracked, captured, and eliminated these bacteria in organs in the treatment group.”

4) Reviewer 4, comment 3: Can the authors label the organs shown in SI Figure 10 and SI Figure 11 for clarity?

Thanks for your suggestions. We have labeled the organs in these figures in the revised support information.

We replaced these figures on page 8, line 2 in the revised supporting information:

Figure 10. Ex-vivo fluorescence imaging of organs from healthy mice 12 h after the administration of D-RAWs.

We replaced these figures on page 8, line 5 in the revised supporting information:

Figure 11 (a, b) D-RAWs (a) and **Lyso700D** (b) real-time fluorescence intensity distribution of major organs and legs in vivo. (c, d) D-RAWs (c) and **Lyso700D** (d) real-time fluorescence intensity statistics of major organs and legs in vivo in an immunodeficient mice epidermal inflammation model.

5) Reviewer 4, comment 8: The original review was not asking to remove irradiance in mW/cm², but to additionally include fluence in J/cm². Irradiance values should be re-inserted.

Thanks for your kind reminder. We've corrected this content in the revised manuscript and revised support information.

Thank you for your kind consideration.

Sincerely,

Xinfu Zhang